

# 1 Variability of TEC and improvement of performance

# 2 of the IRI model over Ethiopia during the high solar

# 3 activity phase

Yekoye Asmare Tariku
Department of Space Science and Research Application Development, Ethiopian Space
Science and Technology Institute, Addis Ababa, Ethiopia
* Corresponding author. Tel. +251912799754
*Email_address:yekoye2002@gmail.com (Yekoye Asmare)*
**Abstract**
This paper discusses the monthly and seasonal variation of the total electron content (TEC) and
the improvement of performance of the IRI model in estimating TEC over Ethiopia during the
solar maximum (2013-2016) phase employing GPS TEC data inferred from the GPS receivers
installed at different regions of Ethiopia. The results reveal that both the measured and modeled
seasonal diurnal VTEC values start increasing at 03:00 UT (06:00 LT) and attain their peak
values (mostly in the time interval of 09:00-13:00 UT or 12:00-16:00 LT). Moreover, both the
arithmetic mean measured and modeled VTEC values, generally, show maximum and minimum
values in the equinoctial and June solstice months, respectively. The results also show that, even
though overestimation of the modeled VTEC has been observed on most of the hours, the model
is generally good to estimate both the monthly and seasonal diurnal hourly VTEC values,
especially in the early morning hours (00:00-03:00 UT or 03:00-06:00 LT). Moreover, the overall



results show that using NeQuick option for the topside electron density is the best option in
estimating the TEC variation. It has also been shown that the model does not show a good
improvement on its performance in estimating both the monthly and seasonal hourly and
arithmetic mean VTEC values. Moreover, the model does not respond to the effects resulting
from storm.
**Key words**: GPS-VTEC; IRI- VTEC; GPS signal, solar maximum
**1. Introduction**

The energy transferred from the sun causes atoms and molecules existing in the

atmosphere to undergo chemical reactions and become ionized (Kelley, 2009). This ionized and
conductive region of Earth's atmosphere, extending from about 50 to 1000 km and possessing
free electrons and positive ions generally in equal numbers in a medium that is electrically
neutral, is termed as ionospher. The existence of these ions (plasma) in the ionosphere results in
the possibility of radio communications over large distance by making use of one or more
ionospheric reflections (Hunsucker and Hargreaves, 2003).

On the other hand, the ionosphere affects the electromagnetic waves that pass through it

by inducing additional transmission time delay (Gao and Liu, 2002). Because of its dispersive
character, electromagnetic signals (such as GPS signals) experience time delay (modulated
codes) and advance (carrier phase) as they propagate through the ionosphere. This delay is
directly proportional to the integral number of electrons in a unit cross-sectional area (usually
referred to as total electron content, TEC) along the signal path extending from the satellite to the
receiver on the ground, and inversely proportional to the square of the frequency of propagation
(Hofmann-Wellenhof et al., 1992; Misra and Enge, 2006). The dispersive ionosphere introduces



a time delay in the 1.57542 GHz (L1) and 1.22760 GHz (L2) simultaneous transmissions from
GPS satellites orbiting at 20,200 km (Hansen et al., 2000). The relative ionospheric delay of the
two signals is proportional to the TEC. Time delay measurements of L1 and L2 frequencies can,
therefore, be converted to TEC along the ray path from the receiver to the satellite (Lanyi and
Roth, 1988). The GPS signals traverses the ionosphere carrying signatures of the dynamic
medium and thus offers opportunities for ionospheric research. As a result, global and regional
maps of ionospheric TEC can be produced using data from the worldwide network of the
International GPS Service (Lanyi and Roth, 1988). The availability of TEC measurements is also
important to the development of ionospheric models such as the International Reference
Ionosphere, IRI (Bilitza, 2001). The International Reference Ionosphere (IRI) is an international
project sponsored by the Committee on Space Research (COSPAR) and the International Union
on Radio Science (URSI).
Using the GPS satellites and the IRI model, there have been so far several researches
conducted globally in connection with the TEC variability and performance of the model over
equatorial and low latitude regions. Based on the findings, the researchers have put their own
views concerning the TEC variation and the prediction performance of different versions of the
model. Ezquer et al. (2014), for instance, noted that IRI 2012 predictions show significant
deviations from experimental values during the period of 2008-2009 for a station placed at the
southern crest of the equatorial anomaly in the American region. Olwendo et al. (2012a) also
noted that seasonal average IRI 2007 TEC values were higher than the GPS-TEC data for the
period of 2009-2011 over different regions in Kenya. In addition, Olwendo et al. (2012b)
reported that the IRI 2007 TEC is too high for all seasons except for the March equinox (where
there seems to be good agreement between observation and model) during the lowest solar





activity phase (2009-2010). The report of Kumar (2016) on the validation of the IRI 2012 models
for the global equatorial region also showed that the IRI 2012 model generally overestimated the
observed VTEC over equatorial regions during the solar minimum year (2009) and solar
maximum (2012) phases. Abdu et al. (1996); Kakinami et al. (2012); Kumar et al. (2015)
attempted to describe the model's capacity to estimate the TEC using different versions of the
model. Asmare et al. (2014) and Tariku, 2015a and Tariku, 2015b  also attempted to see patterns
in both the measured and modeled VTEC variations during the low and high solar activity phases
employing different GPS stations and IRI 2012 model at various regions of Ethiopia. Asmare et
al. (2014), for instance, showed that the model entirely overestimated both monthly and seasonal
VTEC values during phases of low solar activity. In addition, the model performance in
estimating diurnal VTEC variations was found to be better during low solar activity phases than
during high solar activity phases. In addition, the highest and the lowest values of the VTEC are
observed in the equinoctial and the June solstice months, respectively during both the low and
high solar activity phases.

Thus, this study is mainly important to observe the TEC variation and the improvement of

performance of the IRI model in estimating the TEC variation over low latitude African regions
during the high solar activity phase (2013-2016) employing the GPS VTEC data inferred from
different regions of Ethiopia. This is because the TEC is the major parameter that can largely
affect radio wave propagation in the ionosphere. Consequently, for a better radio wave
propagation which can foster Earth-to-space communication, the TEC in the ionosphere has to be
studied and its effects on the signal propagated through the ionosphere must be identified. In
addition, for a better improvement of the IRI model in estimating the variation of TEC, its
performance has to be continuously tested, especially over the equatorial and low latitude





regions, where the dynamics of the ionosphere is very complex. To observe the TEC variation
and improvement of performance of the IRI model in estimating the TEC variation the latest
versions (IRI 2007, IRI 2012 and IRI 2016) with NeQuick option for the topside electron density
during the solar maximum phase have been considered. This is conducted to choose and use the
best version of the model in estimating the TEC variability in some occasions when the GPS
TEC data are scarcely available in the receiver. The prediction performance of the model has
been tested by comparing the modeled TEC values with the GPS-TEC values recorded in the
receivers.

**2. Data description and analysis method**

*2.1. TEC from dual frequency GPS receiver*

As different studies (e.g. Ciraolo et al., 2007; Mannucci et al., 1998) show that the GPS

measurements are used to estimate the TEC along a ray path between a GPS satellite and
receiver on the ground. These GPS measurements can be recorded using either single or dual
frequency GPS receivers. However, to eliminate ionospheric errors in the estimation of TEC dual
frequency receivers are better (Klobuchar, 1996). Moreover, by computing the differential
phases of the code and carrier phase measurements, dual frequency GPS receivers can provide
integral information about the ionosphere and plasma sphere (Ciraolo et al., 2007; Nahavandchi
and Soltanpour, 2008). Hence, in this paper, the GPS-TEC data have been obtained from dual
frequency receiver using pseudo-range and carrier phase measurements. The TEC inferred from
the pseudo-range (P) measurement is given by:





$$TEC_P = \frac{1}{40.3}\left[\frac{f_1^{\,2} f_2^{\,2}}{f_1^{\,2} - f_2^{\,2}}\right](P_2 - P_1).$$
**(1)**

Similarly, the TEC from carrier phase measurement (Φ) is given as
$$TEC_\Phi = \frac{1}{40.3}\left[\frac{f_1^{\,2} f_2^{\,2}}{f_1^{\,2} - f_2^{\,2}}\right](\Phi_1 - \Phi_2),$$
**(2)**

where $f_1$ and $f_2$ can be related with the fundamental frequency, $f_o = 10.23 MHz$
$$f_1 = 154 f_o = 1575.42 MHz,$$
$$f_2 = 120 f_o = 1227.60 MHz.$$
**(3)**

As shown above, by cross correlating the $f_1$ and $f_2$ modulated carrier signals which are
generally assumed to travel along the same path through the ionosphere, the GPS receiver
obtains the time delay of the code and the carrier phase difference. As the TEC obtained from
code pseudo-range measurements is free of ambiguity, but with relatively much noise; and the
TEC obtained from carrier phase measurements has relatively less noise, but is ambiguous,
linearly combining both code pseudo-range and carrier phase measurements for the same satellite
pass is supposed to increase the accuracy of TEC (Klobuchar et al., 1996; Gao and Liu, 2002).
To better characterize the TEC over a given receiver position and see the overall ionization of the
Earth's ionosphere, the slant TEC (STEC) must be converted into equivalent vertical TEC
(VTEC) at the mean ionospheric height, $h_m$=350 km (Mannucci et al., 1998; Norsuzila et al.,
2008, 2009). Hence, the relationship between STEC and VTEC in terms of the zenith angle $\chi'$ at
the Ionospheric Piercing Point (IPP) and the zenith angle $\chi$ at the receiver position can be given
by:
$$VTEC = STEC(\cos \chi'),$$
**(4)**





where,                  .
$$\chi^{'} = \arcsin[\frac{R_e}{R_e + h_m}\sin\chi].$$                  **(5)**

Substituting equation (5) into equation (4) and rearranging, we get
$$VTEC = STEC\left\{\cos\left[\arcsin\left(\frac{R_e}{R_e + h_m}\sin\chi\right)\right]\right\}.$$                  **(6)**

Here, $R_e$ is the radius of the Earth in kilometers.

*2.2. TEC from the International Reference Ionosphere (IRI) model*

The International Reference Ionosphere (IRI) is an international empirical standard

model used for the specification of ionospheric parameters. The model provides average values
of electron density, electron content, electron and ion temperature, and ion composition as a
function of height, location, local time, and sunspot number for magnetically quiet conditions
(Bilitza, 2001; Bilitza et al., 2014; Bilitza et al., 2017). To enhance the capacity of the model,
improvements have been made through the ingestion of all worldwide available data from
ground-based as well as satellite observations. As a result, a new version of the model (IRI 2016)
has been released in 2017 by incorporating some new input parameters that are supposed to
increase its capacity. The IRI 2016 model includes two new model options for the F2-peak
height hmF2 and a better representation of topside ion densities at very low and high solar
activities. The two new options are used in modeling *hmf2* directly and no longer through its
relationship to the propagation factor *M(3000)F2*. Thus, the new model options enable the IRI
2016 model to predict evening peaks that was not possible in the old versions. In addition, the
improvement of the ion composition model in the topside ionosphere can lower the transition





height from close to 1000 km down to almost 600 km in the new version of the model. A number
of smaller changes have also been made concerning the use of solar indices and the speed-up of
the computer program (Bilitza et al., 2017). For a given location, time and date, like the previous
versions of the model, IRI-2016 model provides the monthly averages of ionospheric parameters
(such as TEC) in the altitude range from about 50–2000 km (Bilitza et al., 2017;
http://IRImodel.org.).        For      more      information,       see      the      model      web      site
(http://omniweb.gsfc.nasa.gov/vitmo/iri-vitmo.html) that was accessed for the period of 25-

30/01/2018.


*2.3. Data sources and method of analysis*

The data required for both the experimental and model were obtained from Ethiopian

regions shown in Figure 1 during the solar maximum (2013-2016) phase. Table 1 also shows the
GPS receiver locations used for the study. The raw GPS data for the described station were
obtained     from     the     University     NAVSTAR     Consortium     (UNAVCO     web     site,
http://www.unavco.org/). The data gained from this web site have two forms: observation and
navigation data in which both of them are zipped. To use the data for the desired purpose, the
GG software (GPS-TEC calibrating software) was used to process the required data in five
minutes interval and an elevation cut-off $10^o$ (see Ciraolo, et al., 2007).

To get the required results, the corresponding modeled VTEC values were inferred from the

latest versions of the model (IRI 2007, IRI 2012 and IRI 2016) that include some latest input
parameters which are supposed to improve the capacity of the model in estimating ionospheric
parameters.    The    online    IRI    versions    of    the    model    were    obtained    from
http://omniweb.gsfc.nasa.vitmo.html. To get the VTEC values, the year, date, month, location,



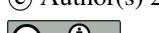

the hour profile, the upper boundary altitude (2000 km), daily sunspot number and F10.7 radio
flux, topside electron density options (NeQuick, IRI01-corr, IRI2001), CCIR for F peak model,
and ABT-2009 for bottomside thickness option were used as the input parameters,.
In order to observe the pattern of the hour-to-hour variability of VTEC, the mean monthly
and seasonal hourly GPS TEC and the corresponding IRI TEC data have been used during the
period of 2013-2016. To see the monthly and seasonal arithmetic mean VTEC variation and the
model performance, the hour-to-hour measured and modeled VTEC values have been
correspondingly added and averaged for the whole days in each month and season. The seasons
could be classified as December solstice (November, December and January), March equinox
(February, March and April), June solstice (May, June and July) and September equinox
(August, September and October). For a better understanding on the performance of the model,
the absolute differences between the monthly and seasonal GPS VTEC and the corresponding
IRI VTEC values have been determined. The differences have been calculated by subtracting the
experimental VTEC values from the model. In order to clearly see the validation of the model,
the absolute differences between the IRI VTEC and GPS VTEC in all the monthly and seasonal
variations were determined. In addition, the percentage differences between the IRI VTEC and
GPS VTEC for the arithmetic monthly and seasonal VTEC variations have also determined.
**3. Results and discussion**
*3.1. Diurnal monthly and seasonal variation of VTEC and performance of the IRI model*

The diurnal monthly and seasonal VTEC variation results are displayed in Figs 2-7. The results
reveal that, almost both the monthly diurnal GPS VTEC and IRI VTEC values start increasing at
03:00 UT (06:00 LT) and attain their peak values in the daytime hours (especially in the time



interval of 09:00-13:00 UT or 12:00-16:00 LT) due to enhancement of ionization during the
described time; while their values start decreasing in the nighttime hours and become minimum
after midnight hours (on average at 03:00 UT or 06:00 LT) as shown in figures 2-7. Moreover, in
some hours, the modeled VTEC values are found to be in a good agreement with the measured
(GPS VTEC) values, especially in the nighttime hours (00:00-03:00 UT or 03:00-06:00 LT).
Moreover, the model is found to underestimate the VTEC values during the daytime hours
(09:00-13:00 UT or 12:00-16:00 LT). The mismodelings observed in both cases may be due to
the difference in the model and experimental slab-thickness as noted by different findings (e.g.
Nigussie et al., 2013; Rios et al., 2007). For instance, Rios et al. (2007) using the IRI 2001
model, showed that IRI predicted slab thickness is higher than the measured values except
between (10:00-14:00 LT) which can attribute to VTEC fluctuations in similar trend. This is
almost consistent with the result determined in this work. Using IRI 2007 model, Nigussie et al.
(2013) also suggested similar possible reason for the discrepancy between the model and the
experimental VTEC values. It could also be resulting from poor estimation of the hmF2 and foF2
from the coefficients, which in turn may result in poor estimation of VTEC by the IRI model
(e.g. Chakraborty et al., 2014; Kumar et al, 2015). The underestimation of the IRI VTEC values
by the GPS VTEC values may also attribute to the enhancement of the plasmaspheric electron
content above 2000 kms during the daytime hours.

Moreover, the maximum peak of both the measured and modeled VTEC values are

generally observed in the equinoctial months; while, the minimum peak values are observed in
the June solstice months (see Fig. 2-7). For instance, over Arba Minch station (see Figs. 2 and 3),
the highest and lowest peak measured monthly VTEC values of about 80 and 40 TECU are
observed in March and July, respectively. Similarly, the highest and lowest peak modeled



seasonal VTEC values of about 55 and 41 TECU are observed in April and July, respectively in
using IRI 2007 model with NeQuick option for the topside electron density.

The seasonal diurnal VTEC values generally follow the pattern of the diurnal monthly

VTEC values with the lowest and highest values being observed at about 03:00 UT (06:00LT)
and in the time interval of about 09:00–13:00 UT (12:00–16:00 LT), respectively (see Fig. 4-7).
In addition, the highest and lowest peak measured seasonal VTEC values of about 80 and 50
TECU are observed in the March equinox and June solstice, respectively. The highest and lowest
peak modeled seasonal VTEC values of about 54 and 43 TECU are observed in the March
equinox and June solstice, respectively when using IRI 2007 model with NeQuick option for the
topside electron density over Arba Minch station (see Fig. 6). In addition, in using IRI-2012
model with IRI-2001 option for the topside electron density, the highest and lowest peak
measured seasonal VTEC values of about 70 and 40 TECU are observed in the March equinox
and June solstice, respectively over Ambo station in 2014. Similarly, the highest and lowest peak
modeled seasonal VTEC values of about 74 and 60 TECU are observed in the March equinox
and June solstice, respectively in 2014 when using the same version of the model (IRI 2012)
with IRI-2001 option (see Fig. 5).

It is known that, in general, electron population in the ionosphere is mainly controlled by

solar photo-ionization and recombination processes (Wu et al., 2004). Thus, for the equinoctial
months, as the subsolar point is around the equator where the east ward electrojet associated
electric field is often largest, it would be speculated that the peak photoelectron abundance and
intense eastward electric field will be set up in the described region. On the contrary, for solstice
months photoelectrons at the equator decrease as the sub solar point moves to higher latitudes.
Moreover, the change of direction of neutral wind may account for the highest VTEC values in

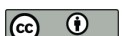



the equinoctial months and lowest values in the June solstice months. A meridional component
of neutral wind blows from the summer to the winter hemisphere that is able to reduce the
ionization crest value during summer solstice as it blows in an opposite direction to the plasma
diffusion process originating from the magnetic equator. Thus, in equinoxes meridional winds
blowing from the equator to polar regions may attribute to a high ionization crest value. Hence, a
seasonal effect on the crest should be expected with the crest maximum at the equinoxes and
minimum in the summer season or June solstice (Bhuyan and Borah, 2007; Wu et al., 2004),
which is consistent with the result of this work.

3.2. *Arithmetic mean of monthly and seasonal variation of VTEC and performance of the IRI*
*model*
To visualize the monthly and seasonal VTEC variations, the arithmetic mean hourly
measured (GPS VTEC) and modeled (IRI VTEC) values obtained during the period of the high
solar activity phase (2013-2016) have been considered. The results are displayed in Figure 8-11.
The results show that both the measured and the modeled mean hourly VTEC have the highest
and lowest values in the equinoctial and June solstice months. For example, the highest and
lowest GPS arithmetic mean monthly VTEC values of about 38 and 18 TECU are observed in
April and July, respectively in the year 2014 over Ambo station. Similarly, the highest and
lowest modeled arithmetic mean monthly VTEC values of about 50 and 35 TECU are observed
in October and July, respectively in the year 2013 when using IRI-2001 option for the topside
electron density (see the left top panel of Fig. 9). The seasonal GPS arithmetic mean VTEC
variation also shows the highest and lowest values of about 37 and 21 TECU in the March
equinox and June solstice, respectively in 2014. In the same way, the highest and lowest seasonal



modeled arithmetic mean VTEC values of about 44 and 39 TECU are observed in the March
equinox and June solstice, respectively in 2013 when using IRI 2012 model with IRI-2001
option (see the bottom left panel of Fig. 8). In addition, the highest and lowest GPS arithmetic
mean monthly VTEC values of about 37 and 17 TECU are observed in March and July,
respectively over Arba Minch station in 2015 (see the top left panels of Fig. 10). Similarly, the
highest and lowest modeled arithmetic mean monthly VTEC values of about 34 and 24 TECU
are observed in April and July, respectively when using IRI 2016 model with NeQuick option for
the topside electron density.   The seasonal GPS arithmetic mean VTEC variation also shows the
highest and lowest values of about 36 and 23 TECU in the March equinox and June solstice,
respectively over Arba Minch station in 2015. In the same way, the highest and lowest seasonal
modeled arithmetic mean VTEC values of about 32 and 24 TECU are observed in the March
equinox and June solstice, respectively when using IRI 2007 model (see the left bottom panels of
Fig. 10). In general, using the IRI 2016 model shows highest overestimation of the VTEC as
compared to others (IRI 2007 and IRI 2012). For instance, the highest monthly and seasonal
deviations of about 25% and 20% are observed between the modeled and corresponding
measured values in September and the June solstice, respectively in using IRI 2016 model (see
the bottom right panels of Fig. 10).

*3.3 Storm Time VTEC Variation and Performance of the IRI Model*
To see the VTEC variation and performance of the IRI model during storm time condition, the
magnetic storm day (with Dst index about -222nT) which occurred on March 17, 2015 as
observed over Arba Minch station was considered.  To better see the effect of the storm on the
GPS VTEC and IRI VTEC, the pattern of the VTEC fluctuations in the initial phase (16/03/2015)



and in the recovery phase (18/03/2015) of the storm was considered. As shown in Fig. 13, the
GPS-VTEC values show significant fluctuation that indicates the occurrence of storm. On the
other hand, the model VTEC values (IRI 2007, IRI 2012 and IRI 2016 VTEC) don't show any
change when the storm model is "on" and "off" (see Figs.13a-13c and Figs.13d-13f). As shown
in the figures, the mode VTEC values in all the three days follow almost similar pattern; they
generally tend to underestimate the VTEC values and remain smooth during the storm. This
shows that the model does not respond to the effects resulting from storm. In addition,
enhancement of GPS TEC is observed as we proceed from the initial to the recovery phase of the
storm. As shown in the figure, a peak VTEC value of about 65 TECU being observed in the
initial phase increases to about 75 TECU in the recovery phase of the storm. This may be
resulted from particle transport and the prompt penetration of high latitude electric field to lower
latitude which travel equator ward with high velocities during the storm (Malik et al., 2010;
Tsurutani et al., 2004; Sobral et al., 2001).

4. Conclusions
Because of the unique geometry of the geomagnetic field near the magnetic equator and low
latitude regions (such as Ethiopia), the signal propagation system through the ionosphere is
largely affected by the accumulation of electrons (TEC). Hence, in this study, the VTEC
variation and the improvement of performance of the IRI model over the equatorial and low
latitude regions has been studied employing the GPS and IRI techniques during the period of
2013-2016. The results reveal that the monthly and seasonal highest peak hourly VTEC values
are mostly observed in the equinoctial months; while the lowest peak values are observed in the
June solstice months. It has also been shown that both the measured and modeled VTEC values



start increasing at about 03:00 UT or 06:00 LT and reach their peak values in the time interval of
about 09:00-13:00 UT or 12:00-16:00 LT. In addition, the maximum and minimum monthly and
seasonal arithmetic mean hourly VTEC values are observed in the equinoctial and June solstice
months, respectively. In addition, though overestimation of the modeled VTEC has been
observed on most of the hours (especially in using IRI 2016 model), the model is generally good
to estimate the diurnal hourly VTEC values mostly just after midnight hours (00:00-03:00 UT or
03:00-06:00 LT). However, the model is found to generally overestimate both the arithmetic
mean of the monthly and seasonal hourly VTEC values in the June solstice and September
equinox months, with the highest overestimation being observed in using IRI 2016 version. On
the other hand, underestimation is observed in the March equinox and December solstice
months, with the highest underestimation being observed in using the same version of the model
(IRI 2016 version). Moreover, the model is found to generally overestimate both the arithmetic
mean of the monthly and seasonal hourly VTEC values, with the highest overestimation being
observed in using IRI-2001 option. The overall results show that using NeQuick option for the
topside electron density is, generally, better than other topside options for TEC estimation by IRI
model. In addition, the model does not show good improvements from version IRI 2007 to IRI
2016 in the TEC estimation over equatorial and low latitude regions. All versions of the model
do not also respond to the effects resulting from storm. Hence, further improvements have to be
made on the model for the betterment of its performance in estimating the VTEC over the
equatorial and low latitude regions.
**Author contribution**
All the required issues for the manuscript are prepared by the corresponding author, Yekoye
**Competing interests**



The corresponding author declares that he has no conflict of interest.
**Acknowledgements**

The data of daily sunspot number, GPS, Dst index and IRI model for this paper are freely
available at: http://www.sidc.be/sunspot-data/,http://facility.unavco.org/data/dai2/app/dai2.,
http://wdc.kugi.kyoto-u.ac.jp/dst_final/201401/index.html and
(http://omniweb.gsfc.nasa.gov/vitmo/iri vitmo.html), respectively. Hence, the author is very
grateful to UNAVCO, NOAA, World Data Center (Kyoto University) and NASA for donating
their free GPS, daily sunspot number, Dst index and online IRI model data, respectively.

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

Figures





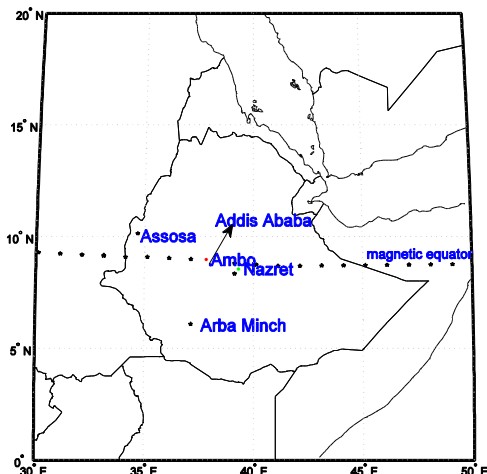


Figure 1: Location of GPS receivers used for the study

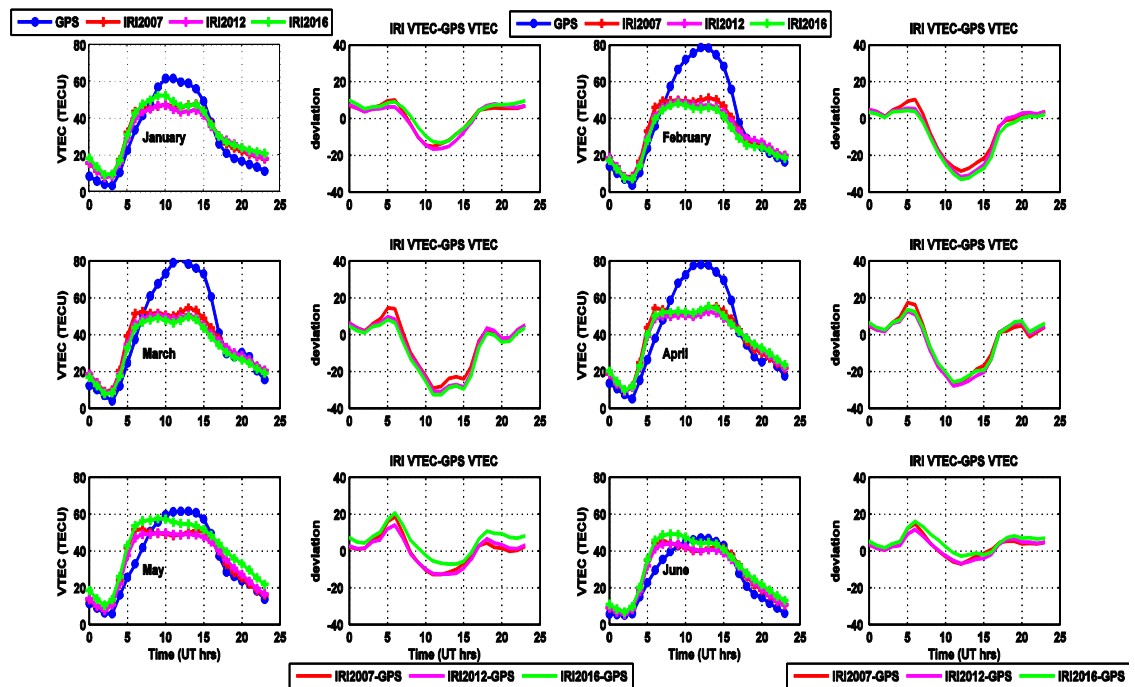


Figure 2: A graph to illustrate diurnal monthly VTEC variation and performance of the IRI
model over Arba Minch station during the period of January-June in 2015





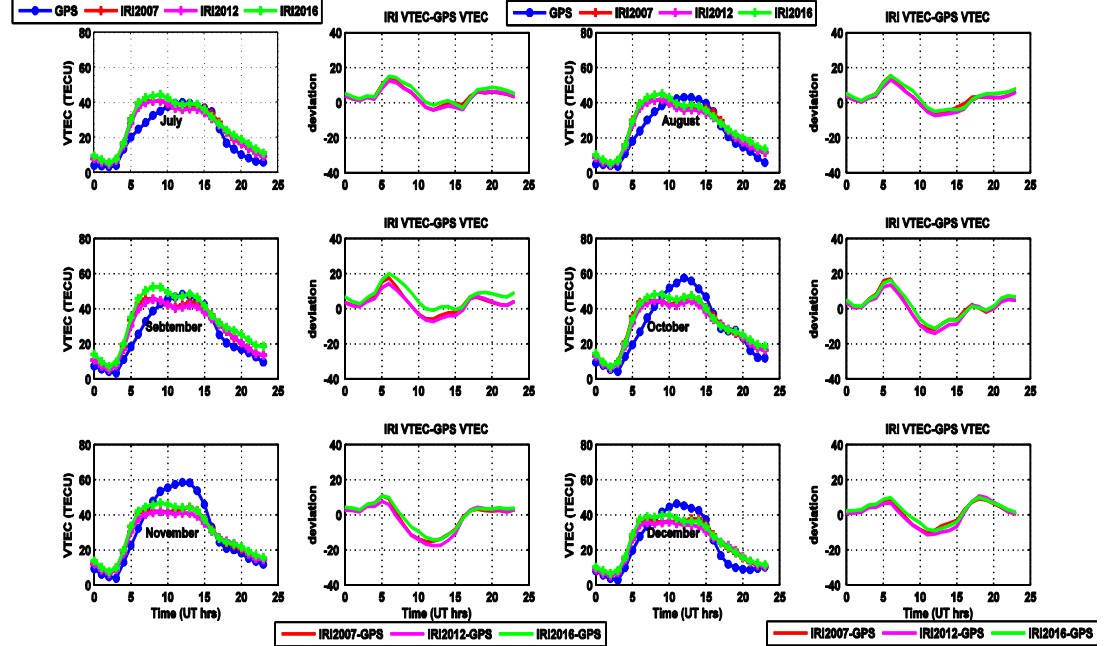


Figure 3: A graph to illustrate diurnal monthly VTEC variation and performance of the IRI

model over Arba Minch station during the period of July-December in 2015

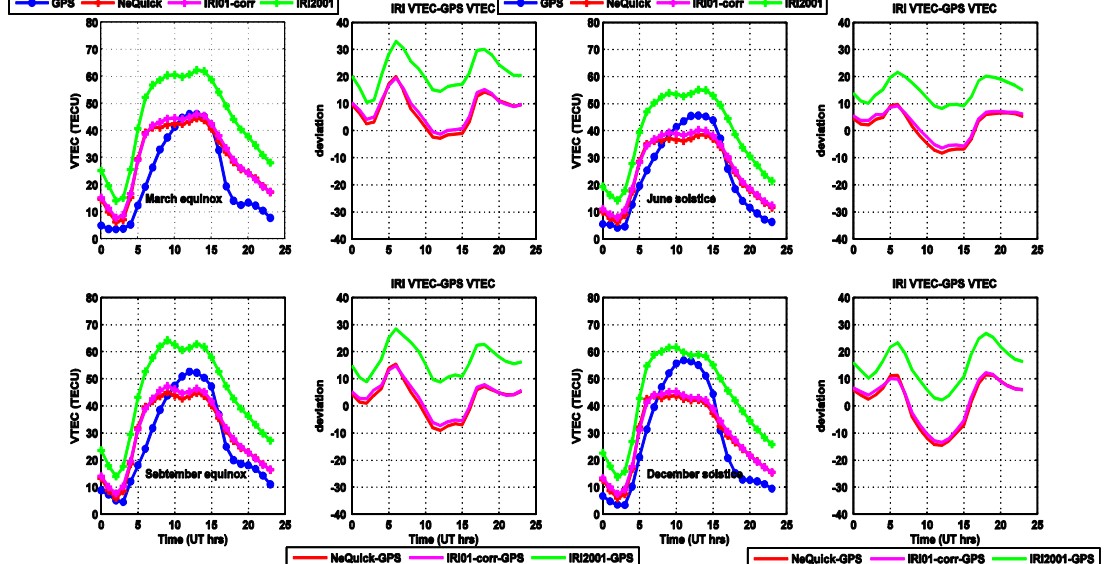





Figure 4: A graph to illustrate diurnal seasonal VTEC variation and performance of the IRI-2012
model over Ambo station during the period of 2013

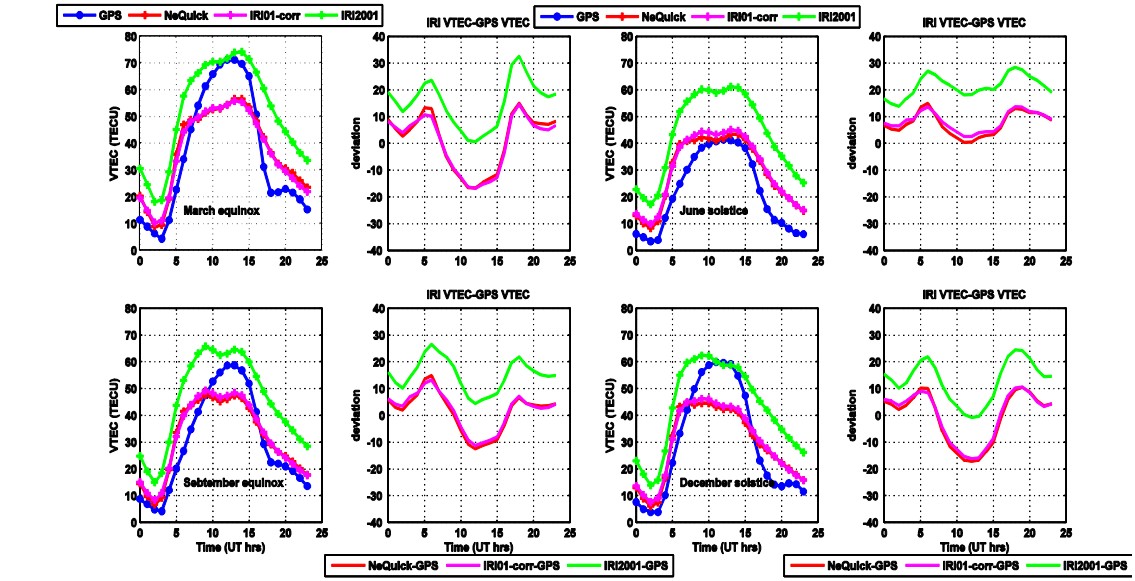


Figure 5: A graph to illustrate diurnal seasonal VTEC variation and performance of the IRI-2012
model over Ambo station during the period of 2014





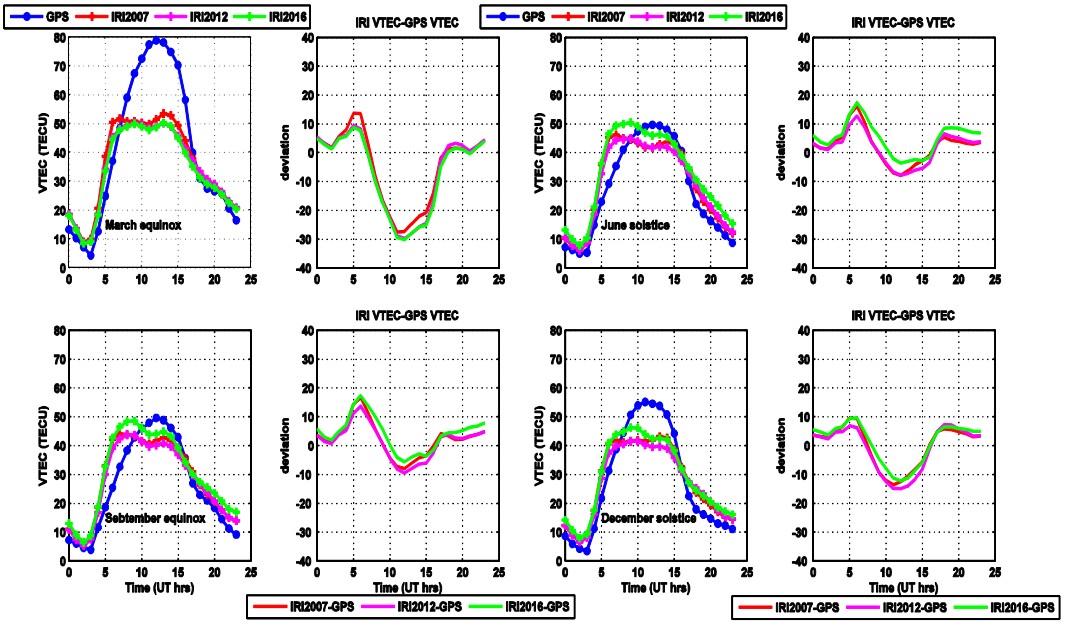



Figure 6: A graph to illustrate diurnal seasonal VTEC variation and performance of the IRI
model over Arba Minch station during the period of 2015

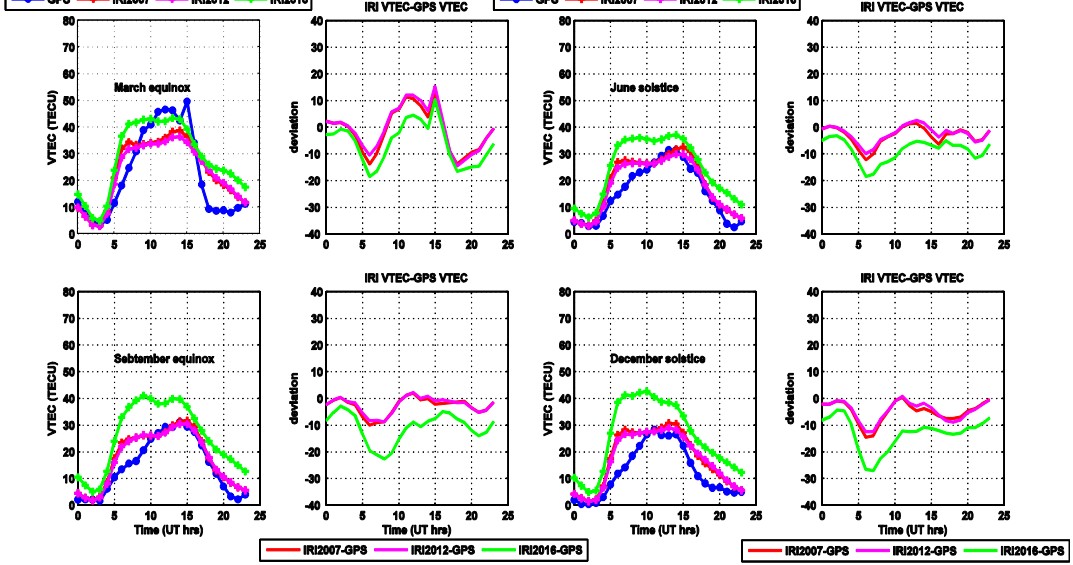





Figure 7: A graph to illustrate diurnal seasonal VTEC variation and performance of the IRI
model over Asosa station during the period of 2016

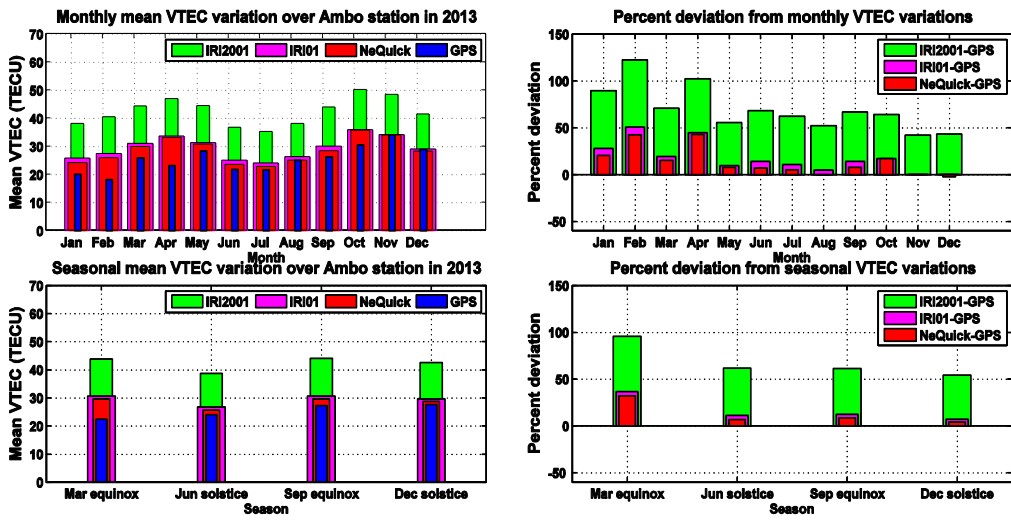


Figure 8: A graph to illustrate the arithmetic mean monthly and seasonal VTEC variation and
performance of the IRI-2012 model over Ambo station during the period of 2013

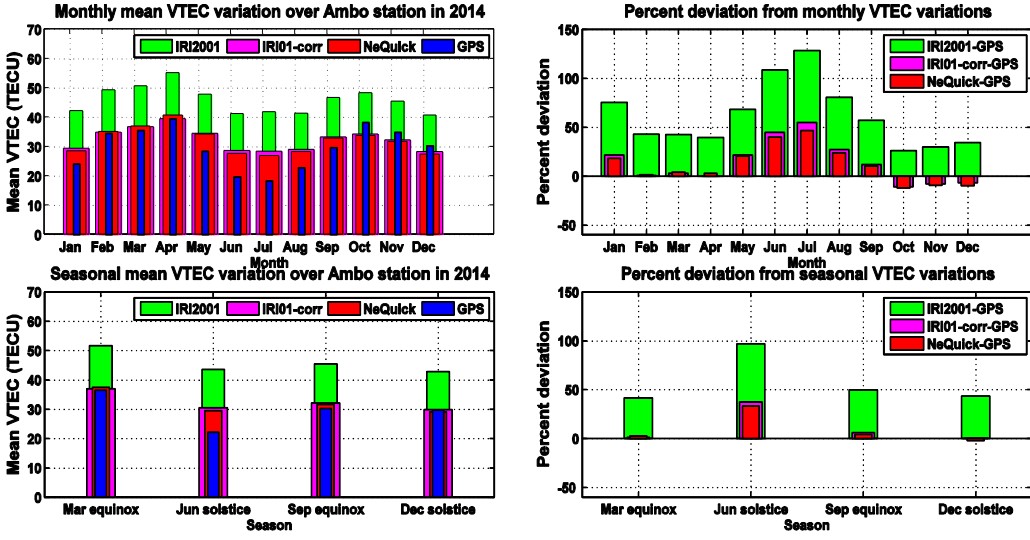




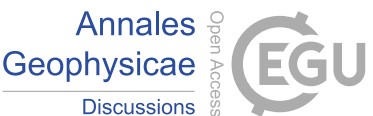

Figure 9: A graph to illustrate the arithmetic mean monthly and seasonal VTEC variation and
performance of the IRI-2012 model over Ambo station during the period of 2014

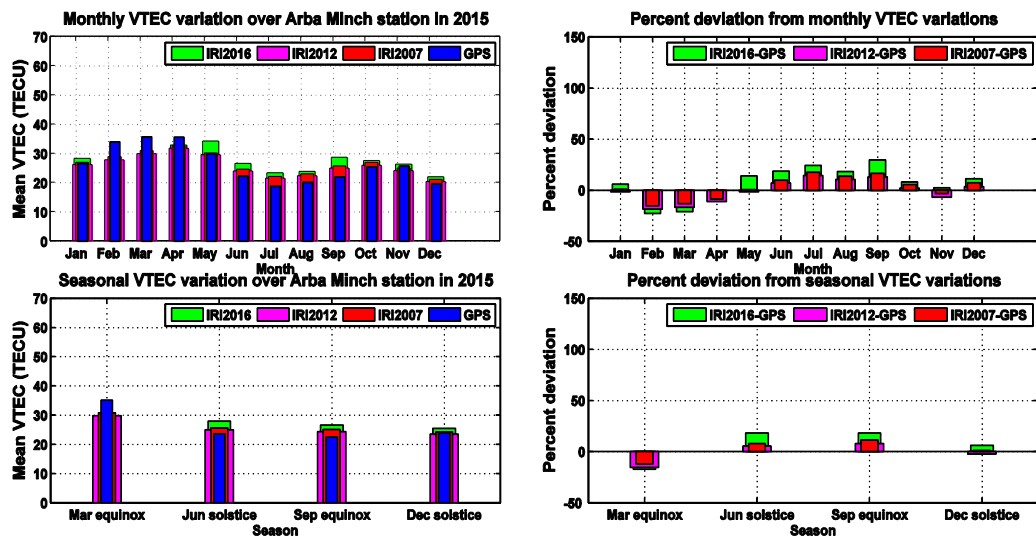



Figure 10: A graph to illustrate the arithmetic mean monthly and seasonal VTEC variation and
performance of the IRI model over Arba Minch station during the period of 2015

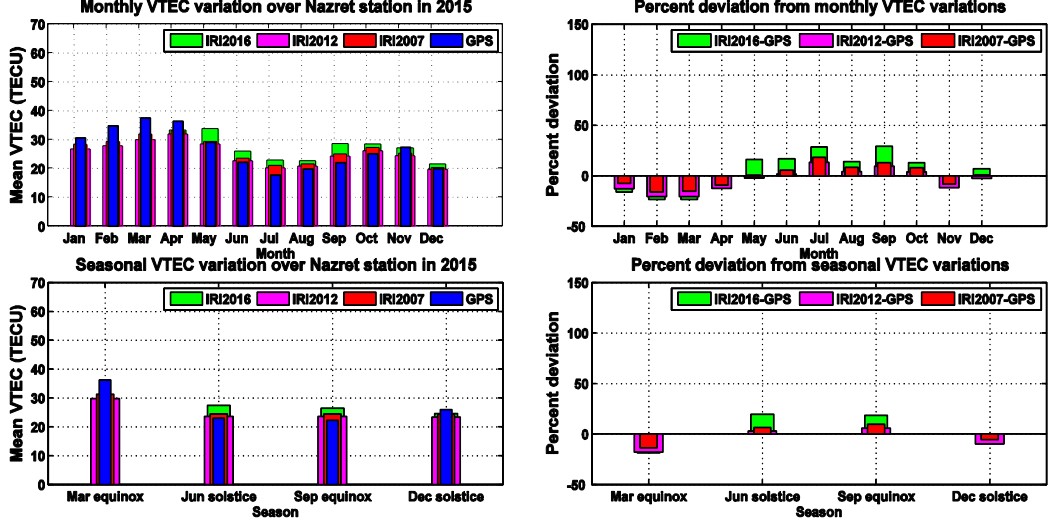






Figure 11: A graph to illustrate the arithmetic mean monthly and seasonal VTEC variation and
performance of the IRI model over Nazret station during the period of 2015

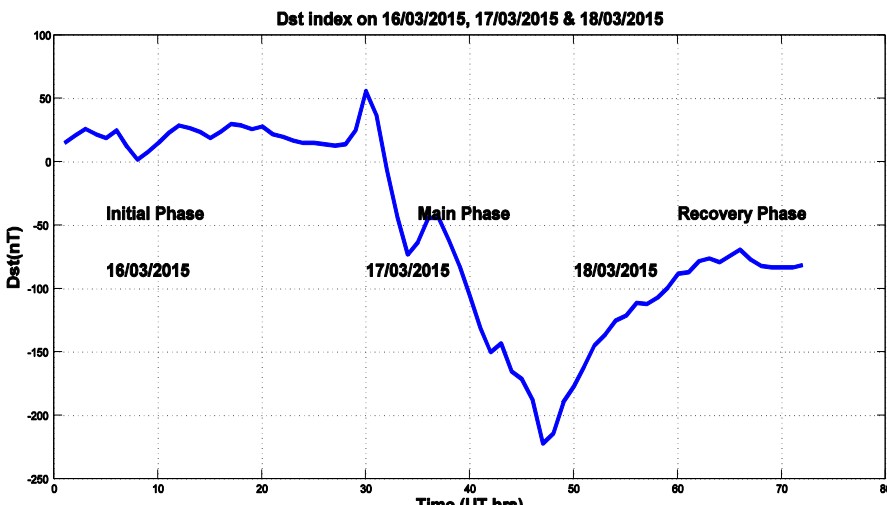



Figure 12: Dst index on 16/03/2015, 17/03/2015, and 18/03/2015 as observed over Arba Minch
station during the period of 2015 (data source for Dst index: World Data Center, Kyoto
University).




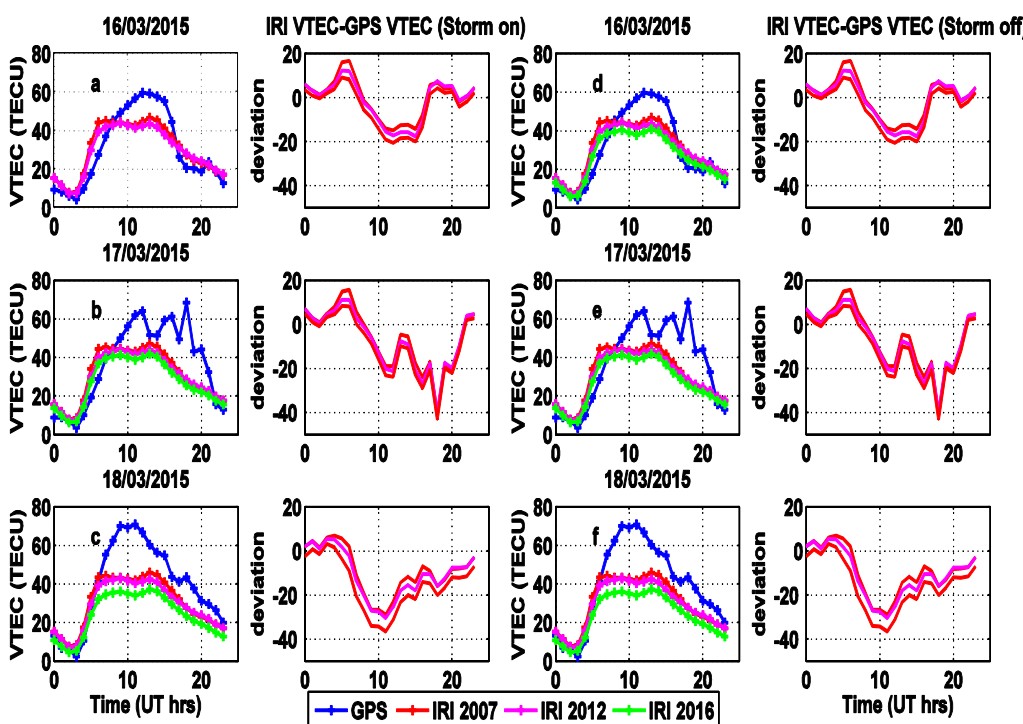

Figure 13: A graph to show the variation of the VTEC and the response of IRI model on storm

time condition which occurred on March 17/2015 as observed over Arba Minch station. Figures

14a–14c and Figures 14d–14f show patterns of the modeled and measured VTEC values when

the storm option is "on" and "off," respectively.

| Station | code | Geographic coordinates Lat. (N), Long. (E) | Geomagnetic coordinates Lat. (N), Long. (E) |
|---|---|---|---|
| Asosa | asos | (10.05,34.55) | (0.56,106.38) |
| Ambo | aboo | (8.97,37.86) | (0.07,109.80) |
| Nazret | nazr | (8.57,39.29) | (-0.08,111.27) |
| Arba MInch | armi | (6.06,37.56) | (-3.08,109.57) |

Table 1: Coordinates of GPS receivers used for the study