# Peer review of "Variability of TEC and improvement of performance 1 of the IRI model over Ethiopia during the high solar 2 activity phase 3 4 Yekoye Asmare Tariku 5 Department of Space Science and Research Application Development, Ethiopian Space 6 Science and Technology Institute, Addis Ababa, Ethiopia 7 8 \* Corresponding author. Tel. +251912799754 *Email address: yekoye2002@gmail.com (Yekoye Asmare)* 9"

_Annales Geophysicae, 2018_

## Referee Comment (RC1) · Anonymous Referee #1 · 3 Jul 2018

**Comments on paper "Variability of TEC and improvement of performance of the IRI model over Ethiopia during the high solar activity phase (angeo_2018-48)" by Tariku (2018).**

July 3, 2018

The paper attempts to discuss the performance of the IRI model over the low latitude region (Ethiopia) in comparison with total electron content derived from Global Navigation Satellite Systems (GNSS), specifically Global Positioning System (GPS) observations. Analysis is done based on computed mean values. The title and the text are not in agreement. In the title there is text "... improvement of performance of the IRI model...", but it is not clear what the IRI model would be improving. If the author is refering to improvement based on different versions of the IRI model, then a sufficient background about the determination of TEC within IRI 2007, IRI 2012 and IRI 2016 should have been provided. In the abstract, the author mentions "The results reveal that both the measured and modelled seasonal diurnal VTEC values start increasing at 03:00 UT (06:00 LT) and attain their peak values (mostly in the time interval of 09:00-13:00 UT or 12:00-16:00 LT)". I do not find this information new to be included in the abstract. The same can be said for the text in lines 22-26.

**1   Summary of the paper**

It is difficult to find new findings in this paper. Most of the results over this region using the same data set have been reported, and even compared to the IRI model. In fact, the same author has reported most of them and so he/she is presenting something that he/she knows has been published. For example in the abstract of Tariku (2015), EPS (paper number 3 below), there is a statement in the abstract " It has been found that the diurnal variability of VTEC has shown minimum values at around 0300 UT (0600 local time (LT)) and maximum values nearly between 1000 and 1300 UT (1300 and 1600 LT) during both the low and the high activity phases": How different is this from the text in lines 14-16 in this submitted manuscript?

Entire subsection 2.1: TEC from dual frequency GPS receiver has been published as many times as the author has published the paper about this topic. In fact equations 1-6 are almost in the same order in the published papers. Examples of the author's papers are provided below where this information appears.

1. TEC prediction of the IRI-2012 model over Ethiopia during the rising phase of solar cycle 24 (2009-2011): Tariku (2015), Earth, Planets and Space, 201567:140, https://doi.org/10.1186/s40623-015-0312-1

2. Comparison of GPS-TEC with IRI-2012 TEC over the African equatorial and low latitude regions during the period of 2012-2013: Tariku (2015), Advances in Space Research, 56, 1677-1685.

3. Patterns of GPS-TEC over low latitude regions (African sector) during the deep solar minimum (2008 to 2009) and solar maximum (2012 to 2013) phases: Tariku (2015), Earth, Planets and Space, 201567:35, https://doi.org/10.1186/s40623-015-0206-2

Other comments are

- In line 171, an elevation threshold of 10 degrees was used. In addition to data being prone multipath errors, I find this low threshold given that the author is performing analysis in low latitude region where electron density gradients are significant.

- Lines 197-202: This text appears in some of the author's papers above and is essentially a repetition or simply some sort of self plagiarism

- Lines 287-304: I think the STORM option is effective for mid-latitudes; Please consult Araujo-Pradere et al., (2002): STORM: An empirical storm-time ionospheric correction model, 2, Validation. Radio Science 37 (5), 1071, doi:10.1029/2002RS002620 and Araujo-Pradere et al., (2004): Time empirical ionospheric correction model (STORM) response in IRI2000 and challenges for empirical modeling in the future. Radio Science 39, RS1S24, doi:10.1029/2002RS002805. Araujo-Pradere et al., (2002): STORM: An empirical storm-time ionospheric correction model, 1, Model description. Radio Science 37 (5), 1070, doi:10.1029/2001RS002467. The author missed these important references about the STORM option model development.

In section 3, subsections 3.1 and 3.2, even the headings are almost the same, with minor editing in the text reported in Tariku (2015), ASR.

Unless the authors show what is different in the submitted paper compared to earlier published papers both by the same author and other existing literature, I find the reported results to have been reported multiple number of times over the same region.
I encourage the author to formulate the objective of the paper and differentiate it from the previously published papers over the same region. Apart from the author's papers, there are other literature sources for this region, e.g, Olwendo et al., (2013); Comparison of GPS TEC variations with IRI-2007 TEC prediction at equatorial latitudes during a low solar activity (2009-2011) phase over the Kenyan region; ASR, Adewale et al., (2012): Solar activity dependence of total electron content derived from GPS observations over Mbarara, ASR; etc.

---

## Referee Comment (RC2) · Anonymous Referee #2 · 4 Jul 2018

Even though the author has a large GNSS data from 2013-2016 over Ethiopia to calculate TEC to compare with many versions of IRI models, the paper should not be accepted since no new contribution, and no substantiated physical explanations are provided. Also the author didn't provide information about how the satellite and receiver biases were determined to calculate the absolute. He only mentioned the Ciraolo et al. (2007) methodology. The tittle induces us to expect an improvement of the IRI model over Ethiopia, what is not provided. The results description from lines 266 to 281 is very repetive and a table should had be used instead. The paper information that TEC is minimum during 03-06 LT and maximum during 12-16 LT is well known.About the storm study it is not expected to get good IRI responses since this model works with

averages and it is very difficult to reproduce storm time behavior. The paper contribution is that there is an overestimation of the modeled VTEC. Small corrections and suggestions are: Line Comments/suggestions/questions 103-105 Correct the phrase 107 Klobuchar et al., 1996 120-124 Improve 150 Is modeling hmf2 referring to using from measurements? 165 sites shown... 177 F10.7 cm solar flux 217 Provide reference 240 Photoionization 242 eastward 244 subsolar 289 Dst index maximum incursion of about... 290 considered (see Figure 12) 291 (16/03/2015), main phase (17/03/2015), 301-304 Which particles and from where? How come penetrated electric field increase TEC? 405-406 Is the reference complete? Figure 1 Ambo and Nazret are overlapped Table 1 Is advisable to inform dip lat instead of geomagnetic coordenates.

---

## Author Comment (AC1) · 10 Jul 2018

Response to reviewers' comments and suggestions The author is very pleased and thankful for editor's and referees' constructive comments and suggestions as I have got the comments, questions, and suggestions helpful in improving my work. Hence, I have presented the replies in the following manner. If the author is referring to improvement based on different versions of the IRI model, then a sufficient background about the determination of TEC within IRI 2007, IRI 2012 and IRI 2016 should have been provided.

Response: the necessary corrections and amendments have been made based on the

suggestion (see lines 60-66).

In the abstract, the author mentions "The results reveal that both the measured and modeled seasonal diurnal VTEC values start increasing at 03:00UT (06:00 LT) and attain their peak values (mostly in the time interval of 09:00-13:00 UT or 12:00-16:00 LT)". I do not find this information new to be included in the abstract. The same can be said for the text in lines 22-26. Response: Some changes have been done (see the abstract section). ———————————————————————————————— It is difficult to find new findings in this paper. Most of the results over this region using the same data set have been reported, and even compared to the IRI model. In fact, the same author has reported most of them and so he/she is presenting something that he/she knows has been published. For example in the abstract of Tariku (2015), EPS (paper number 3 below), there is a statement in the abstract " It has been found that the diurnal variability of VTEC has shown minimum values at around 0300 UT (0600 local time (LT)) and maximum values nearly between 1000 and 1300 UT (1300 and 1600 LT) during both the low and the high activity phases": How different is this from the text in lines 14-16 in this submitted manuscript?

Response: In connection with model validation over low and equatorial regions, of course, a lot of researches have been conducted using the old versions, including IRI 2012 version. However, latest findings that can show the improvement of the model performance from the relatively old to new versions for long lasting period are lacking though the model has been steadily improved and arrived at IRI-2016, which incorporates some new input parameters that did not exist in the previous versions. In addition, only few researches have been conducted to validate the IRI 2016 version of the model over the low and equatorial regions So, to give answer for the question "what is new in the latest versions (especially IRI 2016) of the model in relation to TEC estimation over the region while observing the improvement of the model in general?" this study plays a great role. In short, the study enables to show the improvement of the model from the old to the new version in TEC estimation; and the performance of the most recent version (IRI 2016) in estimating TEC over the region. This is because; validating the new versions of the model enables the model developer to further improve the model. Indeed, here the main purpose is to see the improvement of the IRI model in the estimation of TEC employing IRI 2007, IRI 2012 and IRI 2016 in the same plane using large data for a better accuracy of the results obtained. The past studies might have tested the performance of the model using a single version, either IRI 2007 or IRI 2012. But, few researches have been conducted using IRI 2016. Moreover, as the past studies noted, there are common results obtained in relation to model validation in different version of the model. This shows that the model performance has not been significantly improved. This is one of the basic findings of this study entitled"Assessment of the variability of TEC and improvement of the IRI model. . .". Because the main aim of the study is to show whether the model performance is improved or not. Hence, to further encourage the model developers so that they can significantly improve the model, this study is supposed to give a clear understanding about the improvement of the model performance from the past to the present, especially during the solar maximum phase.

Of course, some modifications have been made (see the revised manuscript to see some new findings that were not discussed in the old manuscript)

Entire subsection 2.1: TEC from dual frequency GPS receiver has been published as many times as the author has published the paper about this topic. In fact equations 1-6 are almost in the same order in the published papers. Examples of the author's papers are provided below where this information appears.

Response: Of course the same equation or similar concepts might have been used for different purposes. For those published papers, the study mainly focuses on the TEC variability or validation of a single version of IRI model (IRI 2012). However, the purpose of the current study is somewhat different from the past studies as it mainly focuses on assessment of the improvement of the IRI model by considering relatively old (IRI 2007) to relatively new model (IRI 2016). The study tries to answer, how the model is showing improvement in TEC estimation from version to version. This is done to make the model developers fill the observed gaps in the model when it is improved from one version to the next one. Here, emphasis has also been given for the most recent version, IRI 2016 as its performance has rarely been observed so far. To see the validation of the model, experimental data have to be used. Hence, in the study GPS data have been used to test the improvement of the model. So, if the GPS data have to be used, GPS related concepts must be raised, including the mechanism of obtaining and utilizing the data. Of course, driving the objectives and the conclusion with that concept (improvement of the performance of the model) may be somewhat lacking. As a result, improvements have been made based on the given suggestions and comments.

* In addition, in the described published papers only sample days' data from each month have been considered, but in this paper, complete monthly and seasonal data have been used to extensively see the variability of TEC and improvement of the model.

Other comments are • In line 171, an elevation threshold of 10 degrees was used. In addition to data being prone multipath errors, I find this low threshold given that the author is performing analysis in low latitude region where electron density gradients are significant.

Response: Multipath effect errors are not that much problems as the receivers are installed at locations far from tall buildings and forests that are supposed to cause poor GPS service (Multipath effect errors). So, calibrating the data at 10o is not a problem.

Lines 197-202: This text appears in some of the author's papers above and is essentially a repetition or simply some sort of self plagiarism Response: some modifications have been made (see the results and discussion section)

Lines 287-304: I think the STORM option is effective for mid-latitudes; Please con- sult Araujo-Pradere et al., (2002): STORM: An empirical storm-time ionospheric correction model, 2, Validation. Radio Science 37 (5), 1071, doi:10.1029/2002RS002620 and

Araujo-Pradere et al., (2004): Time empirical ionospheric correction model (STORM) response in IRI2000 and challenges for empirical modeling in the fu- ture. Radio Science 39, RS1S24, doi:10.1029/2002RS002805. Araujo-Pradere et al., (2002): STORM: An empirical storm-time ionospheric correction model, 1, Model description. Radio Science 37 (5), 1070, doi:10.1029/2001RS002467. The author missed these important references about the STORM option model development.

Response: Here, the main point is to see the performance of the IRI model in esti- mating TEC variation using three versions (IRI 2007, IRI 2012 and IRI 2016) in storm time condition. Because in all versions of the IRI model there is storm time option embedded in the model. So, the objective is to validate the performance of the model during storm time, not to find models used to estimate TEC in storm time condition. The described storm time option models may be tested in other studies. But, now this is beyond the scope of the study. Moreover, it is better to use references related to IRI model rather than using concepts in the described storm time empirical models.

In section 3, subsections 3.1 and 3.2, even the headings are almost the same, with minor editing in the text reported in Tariku (2015), ASR. Response: The pur- pose of this study is different from the past studies as described above Of course, some modifications have also been made here (see the results and discussion section)

Please also note the supplement to this comment: https://www.ann-geophys-discuss.net/angeo-2018-48/angeo-2018-48-AC1-supplement.pdf

**Supplement:**

**Assessment of variability of TEC and improvement of performance of the IRI model over Ethiopia during the high solar activity phase**

Yekoye  Asmare Tariku

Ethiopian Space Science and Technology Institute, Addis Ababa, Ethiopia

* Corresponding author. Tel. +251912799754

*Email_address:yekoye2002@gmail.com (Yekoye Asmare)*

**Abstract**

This paper discusses the monthly and seasonal variation of the total electron content (TEC) and the improvement of performance of the IRI model in estimating TEC over Ethiopia during the solar maximum (2013-2016) phase employing GPS TEC data inferred from the GPS receivers installed at different regions of Ethiopia. **The results reveal that, in the year 2013-2016, the highest peak measured seasonal diurnal VTEC value is observed in the March equinox in 2015 over Arba Minch station**. Moreover, both the arithmetic mean measured and modeled VTEC values, generally, show maximum and minimum values in the equinoctial and June solstice months, respectively in 2014-2015. **However, in 2013, the minimum and maximum arithmetic mean measured values are observed in the March equinox and December solstice, respectively**. The results also show that, even though overestimation of the modeled VTEC has been observed on most of the hours, all versions of the model are generally good to estimate both the monthly and seasonal diurnal hourly VTEC values, especially in the early morning hours (00:00-03:00 UT or 03:00-06:00 LT). **It has also been shown that the IRI 2007 and IRI 2012 versions are generally better when the solar activity decreases; while IRI 2016 is better when the solar activity increases to capture the GPS VTEC values. In addition, the IRI 2012 version with IRI2001 option for the topside electron density shows the highest overestimation of the VTEC as compared to the other options.** All versions of the model do not also able to capture the effects resulting from storm.

**Key words**: GPS-VTEC; IRI- VTEC; GPS signal, solar maximum

**1. Introduction**

The energy transferred from the sun causes atoms and molecules existing in the atmosphere to undergo chemical reactions and become ionized (Kelley, 2009). This ionized and conductive region of Earth's atmosphere, extending from about 50 to 1000 km and possessing free electrons and positive ions generally in equal numbers in a medium that is electrically neutral, is termed as ionospher. The existence of these ions (plasma) in the ionosphere results in the possibility of radio communications over large distance by making use of one or more ionospheric reflections (Hunsucker and Hargreaves, 2003).

On the other hand, the ionosphere affects the electromagnetic waves that pass through it by inducing additional transmission time delay (Gao and Liu, 2002). Because of its dispersive character, electromagnetic signals (such as GPS signals) experience time delay (modulated codes) and advance (carrier phase) as they propagate through the ionosphere. This delay is directly proportional to the integral number of electrons in a unit cross-sectional area (usually referred to as total electron content, TEC) along the signal path extending from the satellite to the receiver on the ground, and inversely proportional to the square of the frequency of propagation (Hofmann-Wellenhof et al., 1992; Misra and Enge, 2006). The dispersive ionosphere introduces a time delay in the 1.57542 GHz (L1) and 1.22760 GHz (L2) simultaneous transmissions from

GPS satellites orbiting at 20,200 km (Hansen et al., 2000). The relative ionospheric delay of the two signals is proportional to the TEC. Time delay measurements of L1 and L2 frequencies can, therefore, be converted to TEC along the ray path from the receiver to the satellite (Lanyi and

Roth, 1988). The GPS signals traverses the ionosphere carrying signatures of the dynamic medium and thus offers opportunities for ionospheric research. As a result, global and regional maps of ionospheric TEC can be produced using data from the worldwide network of the

International GPS Service (Lanyi and Roth, 1988). The availability of TEC measurements is also important to the development of ionospheric models such as the International Reference

Ionosphere, IRI (Bilitza, 2001). The International Reference Ionosphere (IRI) is an international project sponsored by the Committee on Space Research (COSPAR) and the International Union on Radio Science (URSI).

Using the GPS satellites and the IRI model, there have been so far several researches conducted globally in connection with the TEC variability and performance of the model over equatorial and low latitude regions**, especially using IRI 2007 and IRI 2012 versions (e.g.**

**Ezquer et al., 2014; Luhr and Xiong, 2010; Nigussie et al., 2013; Sethi et al., 2011; Olwendo**

**et al., 2012a; Olwendo et al., 2012b**). **Nigussie et al., 2013, for instance, reported that IRI**

**2007 model overestimates the VTEC values over the East African equatorial regions**. **Using**

**IRI 2007, Sethi et al., 2011, also showed that using IRI 2007 model with IRI 2001 option for**

**the topside electron density highly overestimates the VTEC in all seasons and times over**

**low and equatorial Indian regions.** Olwendo et al. (2012a) also noted that seasonal average

IRI 2007 TEC values were higher than the GPS-TEC data for the period of 2009-2011 over different regions in Kenya. In addition, Olwendo et al. (2012b) reported that the IRI 2007 TEC is too high for all seasons except for the March equinox (where there seems to be good agreement between observation and model) during the lowest solar activity phase (2009-2010). Ezquer et al.

(2014), using IRI 2012, noted that IRI 2012 predictions show significant deviations from experimental values during the period of 2008-2009 for a station placed at the southern crest of the equatorial anomaly in the American region. The report of Kumar (2016) on the validation of the IRI 2012 models for the global equatorial region also showed that the IRI 2012 model generally overestimated the observed VTEC over equatorial regions during the solar minimum year (2009) and solar maximum (2012) phases. Asmare et al. (2014) and Tariku, 2015a and

Tariku, 2015b also attempted to see patterns in both the measured and modeled VTEC variations during the low and high solar activity phases employing different GPS stations and IRI 2012

model at various regions of Ethiopia. Asmare et al. (2014), for instance, showed that the IRI

2012 model entirely overestimates both monthly and seasonal VTEC values during phases of low solar activity. In addition, the model performance in estimating diurnal VTEC variations was found to be better during low solar activity phases than during high solar activity phases. In addition, the highest and the lowest values of the VTEC are observed in the equinoctial and the

June solstice months, respectively during both the low and high solar activity phases. Abdu et al.

(1996); Kakinami et al. (2012); Kumar et al. (2015) also attempted to describe the model's capacity to estimate the TEC using different versions of the model. **However, different findings**

**show that the assessment of the improvement of the model performance from the relatively**

**old to new versions for TEC estimation purpose in long lasting period is lacking over low**

**latitude and equatorial regions, such as Ethiopia though the model has been steadily**

**improved and arrived at the most recent version (IRI 2016). Hence, for a better**

**improvement of the IRI model in estimating the variation of TEC, its performance has to be continuously tested, especially over the equatorial and low latitude regions, where the dynamics of the ionosphere is very complex. In addition, there are few researches conducted to test the performance of IRI 2016 model over the region. The model includes some new features that are supposed to enhance its performance in estimating different ionospheric parameters. For instance, the two new model options for the F2-peak height hmF2 and a better representation of topside ion densities at very low and high solar activities enable the model in estimating *hmf2* directly and no longer through its relationship to the propagation factor *M(3000)F2*. As a result, the new model options make the IRI 2016 model estimate evening peaks that was not possible in the old versions.**

Thus, this study is mainly important to observe the TEC variation and the improvement of performance of the IRI model in estimating the TEC variation over low latitude African regions during the high solar activity phase (2013-2016) employing the GPS VTEC data inferred from different regions of Ethiopia. To observe the TEC variation and improvement of performance of the IRI model in estimating the TEC variation the latest versions (IRI 2007, IRI 2012 and IRI 2016) with NeQuick option for the topside electron density during the solar maximum phase have been considered. The prediction performance of the model has been tested by comparing the modeled TEC values with the GPS-TEC values recorded in the receivers.

**2. Data description and analysis method**

*2.1. TEC from dual frequency GPS receiver*

As different studies (such as Ciraolo et al., 2007; Mannucci et al., 1998) show the GPS

measurements are used to estimate the TEC along a ray path between a GPS satellite and receiver on the ground. These GPS measurements can be recorded using either single or dual frequency GPS receivers. However, to eliminate ionospheric errors in the estimation of TEC dual frequency receivers are better (Klobuchar, et al., 1996). Moreover, by computing the differential phases of the code and carrier phase measurements, dual frequency GPS receivers can provide integral information about the ionosphere and plasma sphere (Ciraolo et al., 2007; Nahavandchi and Soltanpour, 2008). Hence, in this paper, the GPS-TEC data have been obtained from dual frequency receiver using pseudo-range and carrier phase measurements. The TEC inferred from the pseudo-range (P) measurement is given by:

$$TEC_P = \frac{1}{40.3}\left[\frac{f_1^2 f_2^2}{f_1^2 - f_2^2}\right](P_2 - P_1).$$
(1)

Similarly, the TEC from carrier phase measurement ($\Phi$) is given as

$$TEC_\Phi = \frac{1}{40.3}\left[\frac{f_1^2 f_2^2}{f_1^2 - f_2^2}\right](\Phi_1 - \Phi_2),$$
(2)

where $f_1$ and $f_2$ can be related with the fundamental frequency, $f_o = 10.23 MHz$

$$f_1 = 154 f_o = 1575.42 MHz,$$
$$f_2 = 120 f_o = 1227.60 MHz.$$
(3)

As shown above, by cross correlating the $f_1$ and $f_2$ modulated carrier signals which are generally assumed to travel along the same path through the ionosphere, the GPS receiver obtains the time delay of the code and the carrier phase difference. The TEC obtained from code pseudo-range measurements is free of ambiguity, but with relatively much noise. On the other hand, the TEC obtained from carrier phase measurements has relatively less noise, but it is ambiguous. Thus, linearly combining both code pseudo-range and carrier phase measurements for the same satellite pass is believed to increase the accuracy of TEC (Ciraolo et al., 2007; Gao and Liu, 2002; Klobuchar et al., 1996). This resultant absolute TEC is the GPS-derived STEC

along the signal from the satellite to the receiver on the ground. To better characterize the TEC

[revised manuscript text omitted]

**3. Results and discussion**

*3.1. Diurnal monthly and seasonal variation of VTEC and assessment of improvements in the*

*performance of the IRI model*

**The results of the variations of the monthly and seasonal hourly VTEC are displayed in**

**Figs 2-7. As observed in the figures, both the measured and modeled VTEC values start**

**decreasing in the nighttime hours (00:00 UT or 03:00 LT) and become minimum after**

**midnight hours (on average at 03:00 UT or 06:00 LT) and start increasing again to attain**

**their peak values in the time interval of about 09:00-13:00 UT or 12:00-16:00 LT).**

Moreover, in some hours, the modeled VTEC values (in all versions) are in a good agreement with the measured (GPS VTEC) values, especially in the nighttime hours (00:00-03:00 UT or

03:00-06:00 LT). On the other hand, all versions of the model tend to underestimate the VTEC

values during the daytime hours (09:00-13:00 UT or 12:00-16:00 LT). **Overestimations are also**

**observed, especially in using IRI 2001 option for IRI 2012 model in 2013-2014 (see Figs. 4**

**and 5) and using IRI 2016 model in 2016 (see Fig. 7). In the year 2013-2016, the highest**

**underestimation (by about 30 TECU) and highest overestimation (by about 20 TECU) are**

[revised manuscript text omitted]

3.2. *Arithmetic mean of monthly and seasonal variations of VTEC and assessment of the*

*improvements in the performance of the IRI model*

**The results of the arithmetic mean monthly and seasonal VTEC variations are given in**

**Figures 8-11**. The results show that both the measured and the modeled arithmetic mean VTEC

have generally the highest and lowest values in the equinoctial and June solstice months. For example, the highest and lowest measured arithmetic mean monthly VTEC values of about 38

and 18 TECU are observed in April and July, respectively in 2014 over Ambo station (see the left top panel of Fig. 9). The seasonal measured arithmetic mean VTEC variation also shows the highest and lowest values of about 37 and 21 TECU in the March equinox and June solstice, respectively in 2014 (see the left bottom panel of Fig. 9). In addition, the highest and lowest seasonal measured VTEC values of about 36 and 23 TECU are observed in the March equinox and June solstice, respectively over Arba Minch station in 2015. The highest and lowest seasonal modeled arithmetic mean VTEC values of about 32 and 24 TECU are also observed in the March equinox and June solstice, respectively when using IRI 2007 version (see the left bottom panels of Fig. 10). On the other hand, the highest and lowest measured monthly VTEC values are observed in November and February, respectively in 2013. Similarly, the highest and lowest measured seasonal VTEC values are observed in the December solstice and March equinox, respectively (see the left top and bottom panels of Fig. 8). But, the highest and lowest modeled arithmetic mean seasonal VTEC values are observed in the March equinox and June, respectively in 2013 when using IRI 2001 option for the topside electron density (see the left top panel of Fig. 8). In **the year 2013-2014, using the IRI 2012 model with IRI2001 option for the**

**topside electron density shows the highest overestimation as compared to NeQuick and**

**IRI01-Corr options. As shown in the Figures (see the right top and bottom panels of Figs. 8**

**and 9), the highest monthly and seasonal overestimations are observed in July (by about**

**130%) and the June solstice (by about 100%) in 2014. On the other hand, the IRI 2012**

**version with NeQuick and IRI01-Corr option relatively gives VTEC having closer values**

**(see Figs 8 and 9). Moreover, the IRI 2016 version shows overestimation of the VTEC as**

**compared to others (IRI 2007 and IRI 2012), especially when the solar activity decreases.**

For instance, the highest monthly and seasonal deviations of about 25% and 20% are observed between the modeled and corresponding measured values in September and the June solstice, respectively when IRI 2016 version is used (see the top and bottom right panels of Fig. 10).

*3.3 Storm Time VTEC variation and performance of the IRI model*

To see the VTEC variation and performance of the IRI model during storm time condition, the magnetic storm day (with Dst index maximum incursion of about -222nT) which occurred on

March 17, 2015 as observed over Arba Minch station was considered (see Fig. 12). To better see the effect of the storm on the GPS VTEC and IRI VTEC, the pattern of the VTEC fluctuations in the initial phase (16/03/2015), main phase (17/03/2015) and in the recovery phase (18/03/2015)

of the storm was considered. As shown in Fig. 13, the GPS-VTEC values show significant fluctuation that indicates the occurrence of storm. On the other hand, the model VTEC values (IRI 2007, IRI 2012 and IRI 2016 VTEC) don't show any change when the storm model is "on"

and "off" (see Figs.13a-13c and Figs.13d-13f). As shown in the figures, the mode VTEC values in all the three days follow almost similar pattern; they generally tend to underestimate the

VTEC values (mostly after 08:00 UT or 11:00 LT) and remain smooth during the storm. This shows that the model does not respond to the effects resulting from storm. **The IRI 2016 VTEC**

**values are also smaller than those of the IRI 2007 and IRI 2012 VTEC values in the initial,**

**main and recovery phase of the storm.** In addition, enhancement of GPS TEC is observed as we proceed from the initial to the recovery phase of the storm. As shown in the figure, a peak

VTEC value of about 65 TECU being observed in the initial phase increases to about 75 TECU

in the recovery phase of the storm. This may be resulted from particle transport and the prompt penetration of high latitude electric fields (PPEFs) to lower latitude which travel equator ward with high velocities during the storm (Malik et al., 2010; Tsurutani et al., 2004; Sobral et al.,

2001**). As the findings show, the dayside ionospheric storms resulting from PPEFs are**

**characterized by transport of near-equatorial plasma to higher altitudes and latitudes,**

**producing a giant plasma fountain. Hence, if the electric field penetrates into the dayside**

**equatorial ionosphere, the plasma is convected toward higher altitudes, forming a giant**

**plasma fountain. At these higher altitudes, the recombination rates are longer than for**

**lower altitudes. On the other hand, solar photoionization at lower altitudes simultaneously**

**continues to occur. This photoionization process will replace the uplifted plasma resulting**

**in an overall increment of TEC.**

4. Conclusions

Because of the unique geometry of the geomagnetic field near the magnetic equator and low latitude regions (such as Ethiopia), the signal propagation system through the ionosphere is largely affected by the accumulation of electrons (TEC). Hence, in this study, the VTEC variation and the improvement of performance of the IRI model over the equatorial and low latitude regions has been studied employing the GPS and IRI techniques during the period of 2013-2016. **The results reveal that the highest and lowest measured and modeled VTEC values are mostly observed in the equinoctial and June solstice months, respectively. However in 2013, the lowest and highest measured seasonal VTEC values are observed in the March equinox and December solstice, respectively. In the year 2013-2016, the maximum seasonal arithmetic mean measured VTEC values are observed in the March equinox except in 2013 in which the minimum and maximum being observed in the March equinox and December solstice, respectively.** In addition, though overestimation of the modeled VTEC has been observed on most of the hours, the model is generally good to estimate the diurnal hourly VTEC values mostly just after midnight hours (00:00-03:00 UT or 03:00-06:00 LT). It has also been shown that the model (IRI 2012) generally overestimates both the arithmetic mean of the monthly and seasonal hourly VTEC values, with the highest overestimation being observed in using IRI2001 option in 2013-2014. The overall results show that using NeQuick option for the topside electron density is generally better than other topside options for TEC estimation by IRI model. In general, the model does not show good improvements from version IRI 2007 to IRI 2016 in the TEC estimation over equatorial and low latitude regions. **However, the IRI 2007 and IRI 2012 versions are generally better to respond to the decrement of the VTEC values when the solar activity decreases; while IRI 2016 version is generally better to capture the measured VTEC values when the solar**

[revised manuscript text omitted]

Venkatesh, K., P. V. S. Rama Rao, P. L. Saranya, D. S. V. V. D. Prasad, and K. Niranjan (2011),

Vertical electron density and topside effective scale height (HT) variations over the Indian equatorial and low latitude stations, Ann. Geophys., 29, 1861–1872, doi:10.5194/angeo-29-

1861-2011.

Wu, C.C., Fry, C.D., Liu, J.Y., Liou, K., Tseng, C.L. (2004); Annual TEC variation in the equatorial anomaly region during the solar minimum: September, 1996-August 1997., J.

Atmos. Terr. Phys., 66:199-207.

Figures

[Figure]

Figure 1: Location of GPS receivers used for the study

[Figure]

Figure 2: A graph to illustrate diurnal monthly VTEC variation and performance of the IRI

model over Arba Minch station during the period of January-June in 2015

[Figure]

Figure 3: A graph to illustrate diurnal monthly VTEC variation and performance of the IRI

model over Arba Minch station during the period of July-December in 2015

[Figure]

Figure 4: A graph to illustrate diurnal seasonal VTEC variation and performance of the IRI-2012

model over Ambo station during the period of 2013

[Figure]

Figure 5: A graph to illustrate diurnal seasonal VTEC variation and performance of the IRI-2012

model over Ambo station during the period of 2014

[Figure]

Figure 6: A graph to illustrate diurnal seasonal VTEC variation and performance of the IRI

model over Arba Minch station during the period of 2015

[Figure]

    Figure 7: A graph to illustrate diurnal seasonal VTEC variation and performance of the IRI

    model over Asosa station during the period of 2016

[Figure]

    Figure 8: A graph to illustrate the arithmetic mean monthly and seasonal VTEC variation and

    performance of the IRI-2012 model over Ambo station during the period of 2013

[Figure]

Figure 9: A graph to illustrate the arithmetic mean monthly and seasonal VTEC variation and performance of the IRI-2012 model over Ambo station during the period of 2014

[Figure]

Figure 10: A graph to illustrate the arithmetic mean monthly and seasonal VTEC variation and performance of the IRI model over Arba Minch station during the period of 2015

[Figure]

Figure 11: A graph to illustrate the arithmetic mean monthly and seasonal VTEC variation and performance of the IRI model over Nazret station during the period of 2015

[Figure]

Figure 12: Dst index on 16/03/2015, 17/03/2015, and 18/03/2015 as observed over Arba Minch station during the period of 2015 (data source for Dst index: World Data Center, Kyoto

University).

[Figure]

Figure 13: A graph to show the variation of the VTEC and the response of IRI model on storm time condition which occurred on March 17/2015 as observed over Arba Minch station. Figures

14a–14c and Figures 14d–14f show patterns of the modeled and measured VTEC values when the storm option is "on" and "off," respectively.

| Station | code | Geographic coordinates Lat. (N), Long. (E) | Geomagnetic coordinates Lat. (N), Long. (E) | Dip angle |
|---|---|---|---|---|
| Asosa | asos | (10.05,34.55) | (0.56,106.38) | 3.2 |
| Ambo | aboo | (8.97,37.86) | (0.07,109.80) | 1.2 |
| Nazret | nazr | (8.57,39.29) | (-0.08,111.27) | 1.19 |
| Arba MInch | armi | (6.06,37.56) | (-3.08,109.57) | -5.7 |

Table 1: Coordinates of GPS receivers used for the study

---

## Author Comment (AC2) · 11 Jul 2018

Response to reviewers' comments and suggestions The author is very pleased and thankful for editor's and referees' constructive comments and suggestions as I have got the comments, questions, and suggestions helpful in improving my work. Hence, I have presented the replies in the following manner. Even though the author has a large GNSS data from 2013-2016 over Ethiopia to calculate TEC to compare with many versions of IRI models, the paper should not be accepted since no new contribution, and no substantiated physical explanations are provided.

Response: In connection with model validation over low and equatorial regions, of course, a lot of researches have been conducted using the old versions, including IRI 2012 version. However, latest findings that can show the improvement of the model performance from the relatively old to new versions for long lasting period are lacking though the model has been steadily improved and arrived at IRI-2016, which incorporates some new input parameters that did not exist in the previous versions. In addition, only few researches have been conducted to validate the IRI 2016 version of the model over the low and equatorial regions So, to give answer for the question "what is new in the latest versions (especially IRI 2016) of the model in relation to TEC estimation over the region while observing the improvement of the model in general?" this study plays a great role. In short, the study enables to show the improvement of the model from the old to the new version in TEC estimation; and the performance of the most recent version (IRI 2016) in estimating TEC over the region. This is because; validating the new versions of the model enables the model developer to further improve the model. Indeed, here the main purpose is to see the improvement of the IRI model in the estimation of TEC employing IRI 2007, IRI 2012 and IRI 2016 in the same plane using large data for a better accuracy of the results obtained. The past studies might have tested the performance of the model using a single version, either IRI 2007 or IRI 2012. But, few researches have been conducted using IRI 2016. Moreover, as the past studies noted, there are common results obtained in relation to model validation in different version of the model. This shows that the model performance has not been significantly improved. This is one of the basic findings of this study entitled"Assessment of the variability of TEC and improvement of the IRI model. . .". Because the main aim of the study is to show whether the model performance is improved or not. Hence, to further encourage the model developers so that they can significantly improve the model, this study is supposed to give a clear understanding about the improvement of the model performance from the past to the present, especially during the solar maximum phase.

Of course, some modifications have been made (see the revised manuscript to see some new findings that were not discussed in the old manuscript)

.Also the author didn0t provide information about how the satellite and receiver biases were determined to calculate the absolute. He only mentioned the Ciraolo et al. (2007) methodology.

Response: It is given in subsection 2.1 (see lines 131-138). The cited reference, Ciraolo et al. (2007 is not necessary.

The tittle induces us to expect an improvement of the IRI model over Ethiopia, what is not provided. The results description from lines 266 to 281 is very repetive and a table should had be used instead. The paper information that TEC is minimum during 03-06 LT and maximum during 12-16 LT is well known.

Response: improvements have been made based on the given suggestions and comments (see the abstract, result and discussion and conclusion sections).

About the storm study it is not expected to get good IRI responses since this model works with averages and it is very difficult to reproduce storm time behavior. The paper contribution is that there is an overestimation of the modeled VTEC.

Response: Here, the main point is to see the performance of the IRI model in estimating TEC variation using three versions (IRI 2007, IRI 2012 and IRI 2016) in storm time condition. Because in all versions of the IRI model there is storm time option embedded in the model. So, the storm option has to be tested whether it fails to capture the VTEC or not, because the objective is to validate the performance of the model during storm time.

Small corrections and suggestions are: Line Comments/suggestions/questions 103-105 Correct the phrase Response: corrected (see lines 114-116)

Klobuchar et al., 1996 Response: corrected (see line 118)

120-124 Improve Response: corrected (see lines 131-138)

Is modeling hmf2 referring to using from measurements? Response: yes Two new

model options for the F2 peak height hmF2, one based on digisonde and one based on radio occultation data. Most significantly, these new options are now modelling hmf2 directly and no longer through its relationship to the propagation factor M(3000)F2.

sites shown... Response: corrected (see line 177)

F10.7 cm solar flux Response: corrected (see line 190)

Provide reference Response: corrected (see lines 238-239)

Photoionization Response: corrected (see line 260) 242 eastward Response: corrected (see line 261) 244 subsolar Response: corrected (see line 264) 289 Dst index maximum incursion of about... Response: corrected (see line 309) 290 considered (see Figure 12) Response: corrected (see line 310)

(16/03/2015), main phase (17/03/2015), Response: corrected (see line 312)

301-304 Which particles and from where? How come penetrated electric field increase TEC? Response: when we say particles, it means plasma (ions and electrons). The particles are transported from the equator towards high altitudes forming a fountain effect. i.e If the electric field penetrates into the dayside equatorial ionosphere, the plasma is convected toward higher altitudes. At these higher altitudes, the recombination rates are considerably longer (hours) than for lower altitudes. Solar photoionization at lower altitudes continues to occur and will replace the uplifted ionosphere/plasma resulting in an overall TEC increase Moreover, additional clarifications are added (see lines 327-334)

405-406 Is the reference complete? Response: corrected (see lines 444-445)

Figure 1 Ambo and Nazret are overlapped Response: corrected (see Figure 1, line 489)

Table 1 Is advisable to inform dip lat instead of geomagnetic coordinates. Response: corrected (see table 1, line 535)
Please also note the supplement to this comment:
https://www.ann-geophys-discuss.net/angeo-2018-48/angeo-2018-48-AC2-supplement.pdf

[Figure]

**Supplement:**

[revised manuscript text omitted]

Using the GPS satellites and the IRI model, there have been so far several researches conducted globally in connection with the TEC variability and performance of the model over equatorial and low latitude regions, especially using IRI 2007 and IRI 2012 versions (e.g. Ezquer et al., 2014; Luhr and Xiong, 2010; Nigussie et al., 2013; Sethi et al., 2011; Olwendo et al., 2012a; Olwendo et al., 2012b). Nigussie et al., 2013, for instance, reported that IRI 2007 model overestimates the VTEC values over the East African equatorial regions. Using IRI 2007, Sethi et al., 2011, also showed that using IRI 2007 model with IRI 2001 option for the topside electron density highly overestimates the VTEC in all seasons and times over low and equatorial Indian regions. Olwendo et al. (2012a) also noted that seasonal average IRI 2007 TEC values were higher than the GPS-TEC data for the period of 2009-2011 over different regions in Kenya. In addition, Olwendo et al. (2012b) reported that the IRI 2007 TEC is too high for all seasons except for the March equinox (where there seems to be good agreement between observation and model) during the lowest solar activity phase (2009-2010). Ezquer et al. (2014), using IRI 2012, noted that IRI 2012 predictions show significant deviations from experimental values during the period of 2008-2009 for a station placed at the southern crest of the equatorial anomaly in the

American region. The report of Kumar (2016) on the validation of the IRI 2012 models for the global equatorial region also showed that the IRI 2012 model generally overestimated the observed VTEC over equatorial regions during the solar minimum year (2009) and solar maximum (2012) phases. Asmare et al. (2014) and Tariku, 2015a and Tariku, 2015b also attempted to see patterns in both the measured and modeled VTEC variations during the low and high solar activity phases employing different GPS stations and IRI 2012 model at various regions of Ethiopia. Asmare et al. (2014), for instance, showed that the IRI 2012 model entirely overestimates both monthly and seasonal VTEC values during phases of low solar activity. In addition, the model performance in estimating diurnal VTEC variations was found to be better during low solar activity phases than during high solar activity phases. In addition, the highest and the lowest values of the VTEC are observed in the equinoctial and the June solstice months, respectively during both the low and high solar activity phases. Abdu et al. (1996); Kakinami et al. (2012); Kumar et al. (2015) also attempted to describe the model's capacity to estimate the

TEC using different versions of the model. **However, different findings show that the**

**assessment of the improvement of the model performance from the relatively old to new**

**versions for TEC estimation purpose in long lasting period is lacking over low latitude and**

**equatorial regions, such as Ethiopia though the model has been steadily improved and**

**arrived at the most recent version (IRI 2016). Hence, for a better improvement of the IRI model in estimating the variation of TEC, its performance has to be continuously tested, especially over the equatorial and low latitude regions, where the dynamics of the ionosphere is very complex. In addition, there are few researches conducted to test the performance of IRI 2016 model over the region. The model includes some new features that are supposed to enhance its performance in estimating different ionospheric parameters. For instance, the two new model options for the F2-peak height hmF2 and a better representation of topside ion densities at very low and high solar activities enable the model in estimating *hmf2* directly and no longer through its relationship to the propagation factor *M(3000)F2*. As a result, the new model options make the IRI 2016 model estimate evening peaks that was not possible in the old versions.**

Thus, this study is mainly important to observe the TEC variation and the improvement of performance of the IRI model in estimating the TEC variation over low latitude African regions during the high solar activity phase (2013-2016) employing the GPS VTEC data inferred from different regions of Ethiopia. To observe the TEC variation and improvement of performance of the IRI model in estimating the TEC variation the latest versions (IRI 2007, IRI 2012 and IRI 2016) with NeQuick option for the topside electron density during the solar maximum phase have been considered. The prediction performance of the model has been tested by comparing the modeled TEC values with the GPS-TEC values recorded in the receivers.

**2. Data description and analysis method**

*2.1. TEC from dual frequency GPS receiver*

As different studies **(such as Ciraolo et al., 2007; Mannucci et al., 1998) show the GPS**

[revised manuscript text omitted]

**The results of the variations of the monthly and seasonal hourly VTEC are displayed in**

**Figs 2-7. As observed in the figures, both the measured and modeled VTEC values start**

**decreasing in the nighttime hours and become minimum after midnight hours (on average**

**at 03:00 UT or 06:00 LT) and start increasing again to attain their peak values in the time**

**interval of about 09:00-13:00 UT or 12:00-16:00 LT).** Moreover, in some hours, the modeled

VTEC values (in all versions) are in a good agreement with the measured (GPS VTEC) values, especially in the nighttime hours (00:00-03:00 UT or 03:00-06:00 LT). On the other hand, all versions of the model tend to underestimate the VTEC values during the daytime hours (09:00-

13:00 UT or 12:00-16:00 LT). **Overestimations are also observed, especially in using IRI**

**2001 option for IRI 2012 model in 2013-2014 (see Figs. 4 and 5) and using IRI 2016 model**

**in 2016 (see Fig. 7). In the year 2013-2016, the highest underestimation (by about 30 TECU)**

**and highest overestimation (by about 20 TECU) are observed in the March equinox in 2015**

**(using IRI 2016 model) and June solstice in 2014 (using IRI 2012 model with IRI 2001**

**option), respectively at about 12:00 UT (15:00 LT). However, IRI 2007 and IRI 2012 are**

**generally better to capture the VTEC values as the solar activity decreases; while, IRI 2016**

**version is generally better when the solar activity increases. Moreover, the IRI 2012 version**

**with NeQuick and IRI01-Corr gives hourly VTEC variation having closer hourly VTEC**

[revised manuscript text omitted]

3.2. *Arithmetic mean of monthly and seasonal variations of VTEC and performance of the IRI*

*model*

**The results of the arithmetic mean monthly and seasonal VTEC variations are given in**

**Figures 8-11**. The results show that both the measured and the modeled arithmetic mean VTEC

have generally the highest and lowest values in the equinoctial and June solstice months. For example, the highest and lowest measured arithmetic mean monthly VTEC values of about 38

and 18 TECU are observed in April and July, respectively in 2014 over Ambo station (see the left top panel of Fig. 9). The seasonal measured arithmetic mean VTEC variation also shows the highest and lowest values of about 37 and 21 TECU in the March equinox and June solstice, respectively in 2014 (see the left bottom panel of Fig. 9). In addition, the highest and lowest seasonal measured VTEC values of about 36 and 23 TECU are observed in the March equinox and June solstice, respectively over Arba Minch station in 2015. The highest and lowest seasonal modeled arithmetic mean VTEC values of about 32 and 24 TECU are also observed in the March equinox and June solstice, respectively when using IRI 2007 version (see the left bottom panels of Fig. 10). On the other hand, the highest and lowest measured monthly VTEC values are observed in November and February, respectively in 2013. Similarly, the highest and lowest measured seasonal VTEC values are observed in the December solstice and March equinox, respectively (see the left top and bottom panels of Fig. 8). But, the highest and lowest modeled arithmetic mean seasonal VTEC values are observed in the March equinox and June, respectively in 2013 when using IRI 2001 option for the topside electron density (see the left top panel of Fig. 8). In **the year 2013-2014, using the IRI 2012 model with IRI2001 option for the**

**topside electron density shows the highest overestimation as compared to NeQuick and**

**IRI01-Corr options. As shown in the Figures (see the right top and bottom panels of Figs. 8**

**and 9), the highest monthly and seasonal overestimations are observed in July (by about**

**130%) and the June solstice (by about 100%) in 2014. On the other hand, the IRI 2012**

**version with NeQuick and IRI01-Corr option relatively gives VTEC having closer values**

**(see Figs 8 and 9). Moreover, the IRI 2016 version shows overestimation of the VTEC as**

**compared to others (IRI 2007 and IRI 2012), especially when the solar activity decreases.**

For instance, the highest monthly and seasonal deviations of about 25% and 20% are observed between the modeled and corresponding measured values in September and the June solstice, respectively when IRI 2016 version is used (see the top and bottom right panels of Fig. 10).

*3.3 Storm Time VTEC variation and performance of the IRI model*

To see the VTEC variation and performance of the IRI model during storm time condition, the magnetic storm day (with **Dst index maximum incursion of about -222nT**) which occurred on

March 17, 2015 as observed over Arba Minch station was considered **(see Fig. 12).** To better see the effect of the storm on the GPS VTEC and IRI VTEC, the pattern of the VTEC fluctuations in

**the initial phase (16/03/2015), main phase (17/03/2015)** and in the recovery phase (18/03/2015) of the storm was considered. As shown in Fig. 13, the GPS-VTEC values show significant fluctuation that indicates the occurrence of storm. On the other hand, the model

VTEC values (IRI 2007, IRI 2012 and IRI 2016 VTEC) don't show any change when the storm model is "on" and "off" (see Figs.13a-13c and Figs.13d-13f). As shown in the figures, the mode

VTEC values in all the three days follow almost similar pattern; they generally tend to underestimate the VTEC values (mostly after 08:00 UT or 11:00 LT) and remain smooth during the storm. This shows that the model does not respond to the effects resulting from storm. **The**

**IRI 2016 VTEC values are also smaller than those of the IRI 2007 and IRI 2012 VTEC**

**values in the initial, main and recovery phase of the storm.** In addition, enhancement of GPS

TEC is observed as we proceed from the initial to the recovery phase of the storm. As shown in the figure, a peak VTEC value of about 65 TECU being observed in the initial phase increases to about 75 TECU in the recovery phase of the storm. This may be resulted from particle transport and the prompt penetration of high latitude electric fields (PPEFs) to lower latitude which travel equator ward with high velocities during the storm (Malik et al., 2010; Tsurutani et al., 2004;

Sobral et al., 2001)**. As the findings show, the dayside ionospheric storms resulting from**

**PPEFs are characterized by transport of near-equatorial plasma to higher altitudes and**

**latitudes, producing a giant plasma fountain. Hence, if the electric field penetrates into the**

**dayside equatorial ionosphere, the plasma is convected toward higher altitudes, forming a**

**giant plasma fountain. At these higher altitudes, the recombination rates are longer than**

**for lower altitudes. On the other hand, solar photoionization at lower altitudes**

**simultaneously continues to occur. This photoionization process will replace the uplifted**

**plasma resulting in an overall increment of TEC.**

4. Conclusions

Because of the unique geometry of the geomagnetic field near the magnetic equator and low latitude regions (such as Ethiopia), the signal propagation system through the ionosphere is largely affected by the accumulation of electrons (TEC). Hence, in this study, the VTEC

variation and the improvement of performance of the IRI model over the equatorial and low latitude regions has been studied employing the GPS and IRI techniques during the period of

2013-2016. **The results reveal that the highest and lowest measured and modeled VTEC**

**values are mostly observed in the equinoctial and June solstice months, respectively.**

**However in 2013, the lowest and highest measured seasonal VTEC values are observed in**

**the March equinox and December solstice, respectively. In the year 2013-2016, the**

**maximum seasonal arithmetic mean measured VTEC values are observed in the March**

**equinox except in 2013 in which the minimum and maximum being observed in the March**

**equinox and December solstice, respectively.** In addition, though overestimation of the modeled VTEC has been observed on most of the hours, the model is generally good to estimate the diurnal hourly VTEC values mostly just after midnight hours (00:00-03:00 UT or 03:00-

06:00 LT). It has also been shown that the model (IRI 2012) generally overestimates both the arithmetic mean of the monthly and seasonal hourly VTEC values, with the highest overestimation being observed in using IRI2001 option in 2013-2014. The overall results show that using NeQuick option for the topside electron density is generally better than other topside options for TEC estimation by IRI model. In general, the model does not show good improvements from version IRI 2007 to IRI 2016 in the TEC estimation over equatorial and low latitude regions. **However, the IRI 2007 and IRI 2012 versions are generally better to**

**respond to the decrement of the VTEC values when the solar activity decreases; while IRI**

**2016 version is generally better to capture the measured VTEC values when the solar**

[revised manuscript text omitted]

Venkatesh, K., P. V. S. Rama Rao, P. L. Saranya, D. S. V. V. D. Prasad, and K. Niranjan (2011),

Vertical electron density and topside effective scale height (HT) variations over the Indian equatorial and low latitude stations, Ann. Geophys., 29, 1861–1872, doi:10.5194/angeo-29-

1861-2011.

Wu, C.C., Fry, C.D., Liu, J.Y., Liou, K., Tseng, C.L. (2004); Annual TEC variation in the equatorial anomaly region during the solar minimum: September, 1996-August 1997., J.

Atmos. Terr. Phys., 66:199-207.

Figures

[Figure]

Figure 1: Location of GPS receivers used for the study

[Figure]

Figure 2: A graph to illustrate diurnal monthly VTEC variation and performance of the IRI

model over Arba Minch station during the period of January-June in 2015

[Figure]

Figure 3: A graph to illustrate diurnal monthly VTEC variation and performance of the IRI

model over Arba Minch station during the period of July-December in 2015

[Figure]

Figure 4: A graph to illustrate diurnal seasonal VTEC variation and performance of the IRI-2012

model over Ambo station during the period of 2013

[Figure]

Figure 5: A graph to illustrate diurnal seasonal VTEC variation and performance of the IRI-2012

model over Ambo station during the period of 2014

[Figure]

Figure 6: A graph to illustrate diurnal seasonal VTEC variation and performance of the IRI

model over Arba Minch station during the period of 2015

[Figure]

Figure 7: A graph to illustrate diurnal seasonal VTEC variation and performance of the IRI

model over Asosa station during the period of 2016

[Figure]

Figure 8: A graph to illustrate the arithmetic mean monthly and seasonal VTEC variation and performance of the IRI-2012 model over Ambo station during the period of 2013

[Figure]

Figure 9: A graph to illustrate the arithmetic mean monthly and seasonal VTEC variation and performance of the IRI-2012 model over Ambo station during the period of 2014

[Figure]

Figure 10: A graph to illustrate the arithmetic mean monthly and seasonal VTEC variation and performance of the IRI model over Arba Minch station during the period of 2015

[Figure]

Figure 11: A graph to illustrate the arithmetic mean monthly and seasonal VTEC variation and performance of the IRI model over Nazret station during the period of 2015

[Figure]

Figure 12: Dst index on 16/03/2015, 17/03/2015, and 18/03/2015 as observed over Arba Minch station during the period of 2015 (data source for Dst index: World Data Center, Kyoto

University).

[Figure]

Figure 13: A graph to show the variation of the VTEC and the response of IRI model on storm time condition which occurred on March 17/2015 as observed over Arba Minch station. Figures

14a–14c and Figures 14d–14f show patterns of the modeled and measured VTEC values when the storm option is "on" and "off," respectively.

| Station | code | Geographic coordinates Lat. (N), Long. (E) | Geomagnetic coordinates Lat. (N), Long. (E) | Dip angle |
|---|---|---|---|---|
| Asosa | asos | (10.05,34.55) | (0.56,106.38) | 3.2 |
| Ambo | aboo | (8.97,37.86) | (0.07,109.80) | 1.2 |
| Nazret | nazr | (8.57,39.29) | (-0.08,111.27) | 1.19 |
| Arba Minch | armi | (6.06,37.56) | (-3.08,109.57) | -5.7 |

Table 1: Coordinates of GPS receivers used for the study

---

## Author Comment (AC3) · 30 Jul 2018

**Assessment of variability of TEC and improvement of performance of the IRI model over Ethiopia during the high solar activity phase**

Yekoye Asmare Tariku

Department of Space Science and Research Application Development, Ethiopian Space Science and Technology Institute, Addis Ababa, Ethiopia

* Corresponding author. Tel. +251912799754

*Email_address:yekoye2002@gmail.com (Yekoye Asmare)*

**Abstract**

This paper discusses the monthly and seasonal variation of the total electron content (TEC) and the improvement of performance of the IRI model in estimating TEC over Ethiopia during the solar maximum (2013-2016) phase employing GPS TEC data inferred from the GPS receivers installed at different regions of Ethiopia. **The results reveal that, in the year 2013-2016, the highest peak measured seasonal diurnal VTEC value is observed in the March equinox in 2015 over Arba Minch station**. Moreover, both the arithmetic mean measured and modeled VTEC values, generally, show maximum and minimum values in the equinoctial and June solstice months, respectively in 2014-2015. **However, in 2013, the minimum and maximum arithmetic mean measured values are observed in the March equinox and December solstice, respectively**. The results also show that, even though overestimation of the modeled VTEC has been observed on most of the hours, all versions of the model are generally good to estimate both the monthly and seasonal diurnal hourly VTEC values, especially in the early morning hours (00:00-03:00 UT or 03:00-06:00 LT). **It has also been shown that the IRI 2007**

**and IRI 2012 versions are generally better when the solar activity decreases; while, IRI**

**2016 is better when the solar activity increases to capture the GPS VTEC values. In**

**addition, the IRI 2012 version with IRI2001 option for the topside electron density shows**

**the highest overestimation of the VTEC as compared to the other options.** All versions of the model do not also able to capture the effects resulting from storm.

**Key words**: GPS-VTEC; IRI- VTEC; GPS signal, solar maximum

**1. Introduction**

The energy transferred from the sun causes atoms and molecules existing in the atmosphere to undergo chemical reactions and become ionized (Kelley, 2009). This ionized and conductive region of Earth's atmosphere, extending from about 50 to 1000 km and possessing free electrons and positive ions generally in equal numbers in a medium that is electrically neutral, is termed as ionosphere. The existence of these ions (plasma) in the ionosphere results in the possibility of radio communications over large distance by making use of one or more ionospheric reflections (Hunsucker and Hargreaves, 2003).

On the other hand, the ionosphere affects the electromagnetic waves that pass through it by inducing additional transmission time delay (Gao and Liu, 2002). Because of its dispersive character, electromagnetic signals (such as GPS signals) experience time delay (modulated codes) and advance (carrier phase) as they propagate through the ionosphere. This delay is directly proportional to the integral number of electrons in a unit cross-sectional area (usually referred to as total electron content, TEC) along the signal path extending from the satellite to the receiver on the ground, and inversely proportional to the square of the frequency of propagation (Hofmann-Wellenhof et al., 1992; Misra and Enge, 2006). The dispersive ionosphere introduces a time delay in the 1.57542 GHz (L1) and 1.22760 GHz (L2) simultaneous transmissions from

GPS satellites orbiting at 20,200 km (Hansen et al., 2000). The relative ionospheric delay of the two signals is proportional to the TEC. Time delay measurements of L1 and L2 frequencies can, therefore, be converted to TEC along the ray path from the receiver to the satellite (Lanyi and

Roth, 1988). The GPS signals traverses the ionosphere carrying signatures of the dynamic medium and thus offers opportunities for ionospheric research. As a result, global and regional maps of ionospheric TEC can be produced using data from the worldwide network of the

International GPS Service (Lanyi and Roth, 1988). The availability of TEC measurements is also important to the development of ionospheric models such as the International Reference

Ionosphere, IRI (Bilitza, 2001). The International Reference Ionosphere (IRI) is an international project sponsored by the Committee on Space Research (COSPAR) and the International Union on Radio Science (URSI).

Using the GPS satellites and the IRI model, there have been so far several researches conducted globally in connection with the TEC variability and performance of the model over equatorial and low latitude regions, especially using IRI 2007 and IRI 2012 versions (e.g. Ezquer et al., 2014; Luhr and Xiong, 2010; Nigussie et al., 2013; Sethi et al., 2011; Olwendo et al.,

2012a; Olwendo et al., 2012b). Nigussie et al., 2013, for instance, reported that IRI 2007 model overestimates the VTEC values over the East African equatorial regions. Using IRI 2007, Sethi et al., 2011, also showed that using IRI 2007 model with IRI 2001 option for the topside electron density highly overestimates the VTEC in all seasons and times over low and equatorial Indian regions. Olwendo et al. (2012a) also noted that seasonal average IRI 2007 TEC values were higher than the GPS-TEC data for the period of 2009-2011 over different regions in Kenya. In addition, Olwendo et al. (2012b) reported that the IRI 2007 TEC is too high for all seasons except for the March equinox (where there seems to be good agreement between observation and model) during the lowest solar activity phase (2009-2010). Ezquer et al. (2014), using IRI 2012, noted that IRI 2012 predictions show significant deviations from experimental values during the period of 2008-2009 for a station placed at the southern crest of the equatorial anomaly in the

American region. The report of Kumar (2016) on the validation of the IRI 2012 models for the global equatorial region also showed that the IRI 2012 model generally overestimated the observed VTEC over equatorial regions during the solar minimum year (2009) and solar maximum (2012) phases. Asmare et al. (2014) and Tariku, 2015a and Tariku, 2015b also attempted to see patterns in both the measured and modeled VTEC variations during the low and high solar activity phases employing different GPS stations and IRI 2012 model at various regions of Ethiopia. Asmare et al. (2014), for instance, showed that the IRI 2012 model entirely overestimates both monthly and seasonal VTEC values during phases of low solar activity. In addition, the model performance in estimating diurnal VTEC variations was found to be better during low solar activity phases than during high solar activity phases. In addition, the highest and the lowest values of the VTEC are observed in the equinoctial and the June solstice months, respectively during both the low and high solar activity phases. Abdu et al. (1996); Kakinami et al. (2012); Kumar et al. (2015) also attempted to describe the model's capacity to estimate the

TEC using different versions of the model. **However, different findings show that the**

**assessment of the improvement of the model performance from the relatively old to new**

**versions for TEC estimation purpose in long lasting period is lacking over low latitude and**

**equatorial regions, such as Ethiopia though the model has been steadily improved and**

**arrived at the most recent version (IRI 2016). Hence, for a better improvement of the IRI model in estimating the variation of TEC, its performance has to be continuously tested, especially over the equatorial and low latitude regions where the dynamics of the ionosphere is very complex. In addition, there are few researches conducted to test the performance of the IRI 2016 model over the region. The model includes some new features that are supposed to enhance its performance in estimating different ionospheric parameters. For instance, the two new model options for the F2-peak height hmF2 and a better representation of topside ion densities at very low and high solar activities enable the model in estimating *hmf2* directly and no longer through its relationship to the propagation factor *M(3000)F2*. As a result, the new model options make the IRI 2016 model estimate evening peaks that was not possible in the old versions.**

Thus, this study is mainly important to observe the TEC variation and the improvement of performance of the IRI model in estimating the TEC variation over low latitude African regions during the high solar activity phase (2013-2016) employing the GPS VTEC data inferred from different regions of Ethiopia. To observe the TEC variation and improvement of performance of the IRI model in estimating the TEC variation, the latest versions (IRI 2007, IRI 2012 and IRI 2016) during the solar maximum phase have been considered. The prediction performance of the model has been tested by comparing the modeled TEC values with the GPS-TEC values recorded in the receivers.

**2. Data description and analysis method**

*2.1. TEC from dual frequency GPS receiver*

As different studies **(such as Ciraolo et al., 2007; Mannucci et al., 1998) show the GPS**

**measurements are used to estimate the TEC along a ray path between a GPS satellite and**

**receiver on the ground.** These GPS measurements can be recorded using either single or dual frequency GPS receivers. However, to eliminate ionospheric errors in the estimation of TEC dual frequency receivers are better **(Klobuchar, et al., 1996).** Moreover, by computing the differential phases of the code and carrier phase measurements, dual frequency GPS receivers can provide integral information about the ionosphere and plasma sphere (Ciraolo et al., 2007;

Nahavandchi and Soltanpour, 2008). Hence, in this paper, the GPS-TEC data have been obtained from dual frequency receiver using pseudo-range and carrier phase measurements. The TEC

inferred from the pseudo-range (P) measurement is given by:

$$TEC_P = \frac{1}{40.3}\left[\frac{f_1^2 f_2^2}{f_1^2 - f_2^2}\right](P_2 - P_1). \tag{1}$$

Similarly, the TEC from carrier phase measurement ($\Phi$) is given as

$$TEC_\Phi = \frac{1}{40.3}\left[\frac{f_1^2 f_2^2}{f_1^2 - f_2^2}\right](\Phi_1 - \Phi_2), \tag{2}$$

where $f_1$ and $f_2$ can be related with the fundamental frequency, $f_o = 10.23 MHz$

$$f_1 = 154 f_o = 1575.42 MHz,$$
$$f_2 = 120 f_o = 1227.60 MHz. \tag{3}$$

As shown above, by cross correlating the $f_1$ and $f_2$ modulated carrier signals which are generally assumed to travel along the same path through the ionosphere, the GPS receiver obtains the time delay of the code and the carrier phase difference. **The TEC obtained from**

**code pseudo-range measurements is free of ambiguity, but with relatively much noise. On**

**the other hand, the TEC obtained from carrier phase measurements has relatively less**

**noise, but it is ambiguous. Thus, linearly combining both code pseudo-range and carrier**

**phase measurements for the same satellite pass is believed to increase the accuracy of TEC**

**(Ciraolo et al., 2007; Gao and Liu, 2002; Klobuchar et al., 1996). This resultant absolute**

**TEC is the GPS-derived STEC along the signal from the satellite to the receiver on the**

[revised manuscript text omitted]

**The results of the variations of the monthly and seasonal hourly VTEC are displayed in**

**Figs. 2-7. As observed in the figures, both the measured and modeled VTEC values start**

**decreasing in the nighttime hours and become minimum after midnight hours (on average**

**at 03:00 UT or 06:00 LT) and start increasing again to attain their peak values in the time**

**interval of about 09:00-13:00 UT or 12:00-16:00 LT).** Moreover, in some hours, the modeled

VTEC values (in all versions) are in a good agreement with the measured (GPS VTEC) values, especially in the nighttime hours (00:00-03:00 UT or 03:00-06:00 LT). On the other hand, all versions of the model tend to underestimate the VTEC values during the daytime hours (09:00-

13:00 UT or 12:00-16:00 LT). **Overestimations are also observed, especially in using IRI**

**2001 option for IRI 2012 model in 2013-2014 (see Figs. 4 and 5) and using IRI 2016 model**

**in 2016 (see Fig. 7). In the year 2013-2016, the highest underestimation (by about 30 TECU) and highest overestimation (by about 20 TECU) are observed in the March equinox in 2015 (using IRI 2016 model) and June solstice in 2014 (using IRI 2012 model with IRI2001 option), respectively at about 12:00 UT (15:00 LT). However, the IRI 2007 and IRI 2012 are generally better to capture the VTEC values as the solar activity decreases; while, IRI 2016 version is generally better when the solar activity increases. Moreover, the IRI 2012 version with NeQuick and IRI01 options gives hourly VTEC variation having closer hourly VTEC values (see Figs. 4 and 5).** The mismodelings observed in both cases may be due to the difference in the model and experimental slab-thickness as noted by different findings (e.g. Nigussie et al., 2013; Rios et al., 2007). For instance, Rios et al. (2007) using the IRI 2001 model, showed that IRI predicted slab thickness is higher than the measured values except between (10:00-14:00 LT) which can attribute to VTEC fluctuations in similar trend. This is almost consistent with the result determined in this work. Using IRI 2007 model, Nigussie et al. (2013) also suggested similar possible reason for the discrepancy between the model and the experimental VTEC values. It could also be resulting from poor estimation of the hmF2 and foF2 from the coefficients, which in turn may result in poor estimation of VTEC by the IRI model (e.g. Chakraborty et al., 2014; Kumar et al, 2015). The underestimation of the IRI VTEC values by the GPS VTEC values may also attribute to the enhancement of the plasmaspheric electron content above 2000 km during the daytime hours **(Coisson et al., 2008; Aggarwal, 2011; Venkatesh et al., 2011).**

Moreover, the maximum peak of both the measured and modeled VTEC values are generally observed in the equinoctial months; while, the minimum peak values are observed in the June solstice months (see Fig. 2-7). For instance, over Arba Minch station (see Figs. 2 and 3), the highest and lowest peak measured monthly VTEC values of about 80 and 40 TECU are observed in March and July, respectively in 2015. Similarly, the highest and lowest peak modeled monthly VTEC values of about 55 and 41 TECU are observed in April and July, respectively in using IRI 2007 model with NeQuick option for the topside electron density. In addition, the highest and lowest peak measured seasonal VTEC values of are observed in the

March equinox and June solstice, respectively in 2015. The highest and lowest peak modeled seasonal VTEC values of about 54 and 43 TECU are also observed in the March equinox and

June solstice, respectively when using IRI 2007 model with NeQuick option for the topside electron density over Arba Minch station (see Fig. 6). In addition, in using IRI 2012 model with

IRI2001 option for the topside electron density, the highest and lowest peak measured seasonal

VTEC values of about 70 and 40 TECU are observed in the March equinox and June solstice, respectively over Ambo station in 2014. Similarly, the highest and lowest peak modeled seasonal

VTEC values of about 74 and 60 TECU are observed in the March equinox and June solstice, respectively in 2014 when using the same version of the model (IRI 2012) with IRI2001 option (see Fig. 5). **The overall results show that, in the year 2013-2016, the highest peak measured**

**VTEC values of about 80 TECU is observed in the March equinox in 2015.**

It is known that, in general, electron population in the ionosphere is mainly controlled by solar **photoionization** and recombination processes (Wu et al., 2004). Thus, for the equinoctial months, as the subsolar point is around the equator where the **eastward** electrojet associated electric field is often largest, it would be speculated that the peak photoelectron abundance and intense eastward electric field will be set up in the described region. On the contrary, for solstice months photoelectrons at the equator decrease as the **subsolar** point moves to higher latitudes.

Moreover, the change of direction of neutral wind may account for the highest VTEC values in the equinoctial months and lowest values in the June solstice months. A meridional component of neutral wind blows from the summer to the winter hemisphere that is able to reduce the ionization crest value during summer solstice as it blows in an opposite direction to the plasma diffusion process originating from the magnetic equator. Thus, in equinoxes meridional winds blowing from the equator to polar regions may attribute to a high ionization crest value. Hence, a seasonal effect on the crest should be expected with the crest maximum at the equinoxes and minimum in the summer season or June solstice (Bhuyan and Borah, 2007; Wu et al., 2004), which is consistent with the result of this work.

3.2. *Arithmetic mean of monthly and seasonal variations of VTEC and performance of the IRI*

*model*

**The results of the arithmetic mean monthly and seasonal VTEC variations are given in**

**Figures 8-11**. The results show that both the measured and the modeled arithmetic mean VTEC

have generally the highest and lowest values in the equinoctial and June solstice months. For example, the highest and lowest measured arithmetic mean monthly VTEC values of about 38

and 18 TECU are observed in April and July, respectively in 2014 over Ambo station (see the left top panel of Fig. 9). The seasonal measured arithmetic mean VTEC variation also shows the highest and lowest values of about 37 and 21 TECU in the March equinox and June solstice, respectively in 2014 (see the left bottom panel of Fig. 9). In addition, the highest and lowest seasonal measured VTEC values of about 36 and 23 TECU are observed in the March equinox and June solstice, respectively over Arba Minch station in 2015. The highest and lowest seasonal modeled arithmetic mean VTEC values of about 32 and 24 TECU are also observed in the March equinox and June solstice, respectively when using IRI 2007 version (see the left bottom panels of Fig. 10). On the other hand, the highest and lowest measured monthly VTEC values are observed in November and February, respectively in 2013. Similarly, the highest and lowest measured seasonal VTEC values are observed in the December solstice and March equinox, respectively (see the left top and bottom panels of Fig. 8). But, the highest and lowest modeled arithmetic mean seasonal VTEC values are observed in the March equinox and June, respectively in 2013 when using IRI 2001 option for the topside electron density (see the left top panel of Fig. 8). In **the year 2013-2014, using the IRI 2012 model with IRI2001 option for the**

**topside electron density shows the highest overestimation as compared to NeQuick and**

**IRI01 options. As shown in the Figures (see the right top and bottom panels of Figs. 8 and**

**9), the highest monthly and seasonal overestimations are observed in July (by about 130%)**

**and the June solstice (by about 100%) in 2014. On the other hand, the IRI 2012 version**

**with NeQuick and IRI01 options relatively gives VTEC having closer values (see Figs 8 and**

**9). Moreover, the IRI 2016 version shows overestimation of the VTEC as compared to**

**others (IRI 2007 and IRI 2012), especially when the solar activity decreases.** For instance, the highest monthly and seasonal deviations of about 25% and 20% are observed between the modeled and corresponding measured values in September and the June solstice, respectively when IRI 2016 version is used (see the top and bottom right panels of Fig. 10).

*3.3 Storm Time VTEC variation and performance of the IRI model*

To see the VTEC variation and performance of the IRI model during storm time condition, the magnetic storm day (with **Dst index maximum incursion of about -222nT**) which occurred on

March 17, 2015 as observed over Arba Minch station was considered **(see Fig. 12). As shown in**

**the figure (see Figure 12), the storm started with a sudden impulse/sudden storm**

**commencement at 04:45 UTC on 17 March. This sudden impulse represents a sharp change in how the solar wind was driving space weather conditions at the Earth, including space weather conditions in the ionosphere. Thus, the sudden impulse acts as a shock to the magnetosphere-ionosphere system**. **As a result**, **to better see the effect of the storm on the VTEC, the patterns of the VTEC fluctuations during conditions prior to the onset of the storm (16/03/2015) and the recovery phase (18/03/2015) of the storm were also considered.** As shown in Fig. 13, the GPS-VTEC values show significant fluctuation that indicates the occurrence of storm. On the other hand, the model VTEC values (IRI 2007, IRI 2012 and IRI 2016 VTEC) don't show any change when the storm model is "on" and "off" (see Figs.13a-13c and Figs.13d-13f). As shown in the figures, the mode VTEC values in all the three days follow almost similar pattern; they generally tend to underestimate the VTEC values (mostly after 08:00 UT or 11:00 LT) and remain smooth during the storm. This shows that the model does not respond to the effects resulting from storm. **The IRI 2016 VTEC values are also smaller than those of the IRI 2007 and IRI 2012 VTEC values during conditions prior to the onset of the storm, main and recovery phase of the storm.** In addition, enhancement of GPS TEC is observed as we proceed from the initial to the recovery phase of the storm. As shown in the figure, a peak VTEC value of about 65 TECU being observed during conditions prior to the onset of the storm increases to about 75 TECU in the recovery phase of the storm. This may be resulting from particle transport and the prompt penetration of high latitude electric fields (PPEFs) to lower latitude which travel equator ward with high velocities during the storm (Malik et al., 2010; Sobral et al., 2001; Tsurutani et al., 2004)**. As the findings show, the dayside ionospheric storms resulting from PPEFs are characterized by transport of near-equatorial plasma to higher altitudes and latitudes, producing a giant plasma fountain. Hence, if the**

**electric field penetrates into the dayside equatorial ionosphere, the plasma is convected toward higher altitudes, forming a giant plasma fountain. At these higher altitudes, the recombination rates are longer than for lower altitudes. On the other hand, solar photoionization at lower altitudes simultaneously continues to occur. This photoionization process will replace the uplifted plasma resulting in an overall increment of TEC.**

4. Conclusions

Because of the unique geometry of the geomagnetic field near the magnetic equator and low latitude regions (such as Ethiopia), the signal propagation system through the ionosphere is largely affected by the accumulation of electrons (TEC). Hence, in this study, the VTEC variation and the improvement of performance of the IRI model over the equatorial and low latitude regions has been studied employing the GPS and IRI techniques during the period of 2013-2016. **The results reveal that, in the year 2013-2016, the maximum seasonal arithmetic mean measured VTEC values are observed in the March equinox except in 2013 in which the minimum and maximum being observed in the March equinox and December solstice, respectively.** In addition, though overestimation of the modeled VTEC has been observed on most of the hours, the model is generally good to estimate the diurnal hourly VTEC values mostly just after midnight hours (00:00-03:00 UT or 03:00-06:00 LT). It has also been shown that the IRI 2012 version of the model generally overestimates both the arithmetic mean of the monthly and seasonal hourly VTEC values, with the highest overestimation being observed in using IRI2001 option in 2013-2014. In general, the model does not show good improvements from version IRI 2007 to IRI 2016 in the TEC estimation over equatorial and low latitude regions. **However, the IRI 2007 and IRI 2012 versions are generally better to respond to the**

**decrement of the VTEC values when the solar activity decreases; while IRI 2016 version is**

**generally better to capture the measured VTEC values when the solar activity increases.**

**Moreover,** all versions of the model do not respond to the effects resulting from storm. Hence, further improvements have to be made on the model for the betterment of its performance in estimating the VTEC over the equatorial and low latitude regions.

**Author contribution**

All the required issues for the manuscript are prepared by the corresponding author, Yekoye

**Competing interests**

The corresponding author declares that he has no conflict of interest.

**Acknowledgements**

The data of daily sunspot number, GPS, Dst index and IRI model for this paper are freely available at: http://www.sidc.be/sunspot-data/,http://facility.unavco.org/data/dai2/app/dai2., http://wdc.kugi.kyoto-u.ac.jp/dst_final/201401/index.html and (http://omniweb.gsfc.nasa.gov/vitmo/iri vitmo.html), respectively. Hence, the author is very grateful to UNAVCO, NOAA, World Data Center (Kyoto University) and NASA for donating their free GPS, daily sunspot number, Dst index and online IRI model data, respectively.

Figures

[Figure]

Figure 1: Location of GPS receivers used for the study

[Figure]

Figure 2: A graph to illustrate diurnal monthly VTEC variation and performance of the IRI

model over Arba Minch station during the period of January-June in 2015

[Figure]

Figure 3: A graph to illustrate diurnal monthly VTEC variation and performance of the IRI

model over Arba Minch station during the period of July-December in 2015

[Figure]

Figure 4: A graph to illustrate diurnal seasonal VTEC variation and performance of the IRI-2012

model over Ambo station during the period of 2013

[Figure]

Figure 5: A graph to illustrate diurnal seasonal VTEC variation and performance of the IRI-2012

model over Ambo station during the period of 2014

[Figure]

Figure 6: A graph to illustrate diurnal seasonal VTEC variation and performance of the IRI

model over Arba Minch station during the period of 2015

[Figure]

 Figure 7: A graph to illustrate diurnal seasonal VTEC variation and performance of the IRI

 model over Asosa station during the period of 2016

[Figure]

 Figure 8: A graph to illustrate the arithmetic mean monthly and seasonal VTEC variation and

 performance of the IRI-2012 model over Ambo station during the period of 2013

[Figure]

Figure 9: A graph to illustrate the arithmetic mean monthly and seasonal VTEC variation and performance of the IRI-2012 model over Ambo station during the period of 2014

[Figure]

Figure 10: A graph to illustrate the arithmetic mean monthly and seasonal VTEC variation and performance of the IRI model over Arba Minch station during the period of 2015

[Figure]

Figure 11: A graph to illustrate the arithmetic mean monthly and seasonal VTEC variation and performance of the IRI model over Nazret station during the period of 2015

[Figure]

Figure 12: Dst index on 16/03/2015, 17/03/2015, and 18/03/2015 as observed over Arba Minch station during the period of 2015 (data source for Dst index: World Data Center, Kyoto

University).

[Figure]

Figure 13: A graph to show the variation of the VTEC and the response of IRI model on storm time condition which occurred on March 17/2015 as observed over Arba Minch station. Figures

14a–14c and Figures 14d–14f show patterns of the modeled and measured VTEC values when the storm option is "on" and "off," respectively.

| Station | code | Geographic coordinates Lat. (N), Long. (E) | Geomagnetic coordinates Lat. (N), Long. (E) | Dip angle |
|---------|------|------|------|------|
| Asosa | asos | (10.05,34.55) | (0.56,106.38) | 3.2 |
| Ambo | aboo | (8.97,37.86) | (0.07,109.80) | 1.2 |
| Nazret | nazr | (8.57,39.29) | (-0.08,111.27) | 1.19 |
| Arba Minch | armi | (6.06,37.56) | (-3.08,109.57) | -5.7 |

Table 1: Coordinates of GPS receivers used for the study